# Mechanistic origins of temperature scaling in the early embryonic cell cycle

Jan Rombouts [1,2,5,6], Franco Tavella [3,6], Alexandra Vandervelde[1], Connie Phong[4], James E. Ferrell Jr. [4], Qiong Yang [3,6] ✉ & Lendert Gelens [1,6] ✉

Temperature strongly influences physiological and ecological processes, particularly in ectotherms. While complex physiological rates often follow Arrhenius-like scaling, originally formulated for single reactions, the underlying reasons remain unclear. Here, we examine temperature scaling of the early embryonic cell cycle across six ectothermic species, including *Xenopus*, *Danio rerio*, *Caenorhabditis*, and *Drosophila*. We find remarkably consistent apparent activation energies ($75 \pm 7$ kJ/mol), corresponding to a $Q_{10}$ of 2.8 at 20°C. Computational modeling shows that both biphasic scaling in key cell cycle components and mismatches in activation energies across partially rate-determining enzymes can explain the observed approximate Arrhenius behavior and its breakdown at temperature extremes. Experimental data from cycling *Xenopus* extracts and in vitro assays of individual regulators support both mechanisms. These findings provide mechanistic insights into the biochemical basis of temperature sensitivity and the failure of biological processes at thermal limits.

Living organisms are continually influenced by their environment, and embryos are particularly sensitive to environmental changes. Even subtle perturbations during critical developmental windows can significantly impact embryo viability, as well as embryonic and post-embryonic performance[1]. A key aspect of the environment is temperature: changes in temperature can profoundly influence embryonic development, influencing the speeds of biochemical reactions and affecting the overall physiology, behavior, and fitness of the organism[2–6].

Ectotherms, in particular, rely critically on the ambient temperature as they have minimal ability to generate heat internally. Each ectothermic species has a specific temperature range associated with its geographic distribution on the planet[7,8]. While adult stages of ectotherms can adopt various physiological and behavioral strategies to maintain optimal temperatures, such as taking shelter when it is too hot[6], embryos possess limited mechanisms to cope with

environmental challenges, making them the most vulnerable life stage to environmental stress[9,10]. Temperature plays a pivotal role in determining the fertilization rate of eggs, the growth and survival of embryos, and in certain cases, even the gender of offspring, as observed in many turtle species and all crocodiles[11]. Assessing the thermal impact on and sensitivity of embryonic development across a range of temperatures provides essential insights into species' responses and vulnerability to the challenges posed by global warming[8,9,12]. Indeed, the impact of global warming is already evident in certain sea turtle species, where a diminishing number of male offspring is observed[13].

But even without this shifting landscape and the challenges of their changing ecosystems, ectotherms face the daunting challenge of needing to have their biochemistry function reliably over a wide range of temperatures. Given that complex metabolic networks, signaling

[1]Laboratory of Dynamics in Biological Systems, Department of Cellular and Molecular Medicine, KU Leuven, Herestraat, 49, Leuven, Belgium. [2]Cell Biology and Biophysics Unit and Developmental Biology Unit, European Molecular Biology Laboratory (EMBL), Heidelberg, Germany. [3]Department of Physics / Biophysics, University of Michigan, Ann Arbor, MI, USA. [4]Department of Chemical and Systems Biology, Stanford University School of Medicine, Stanford, CA, USA. [5]Present address: Unit of Theoretical Chronobiology, Université Libre de Bruxelles, Brussels, Belgium. [6]These authors contributed equally: Jan Rombouts, Franco Tavella, Qiong Yang, Lendert Gelens. ✉e-mail: qiongy@umich.edu; lendert.gelens@kuleuven.be

systems, and developmental processes may involve dozens of enzymes, the question arises as to how much variation in the individual enzymes' temperature scaling can be tolerated before the system fails.

The influence of temperature on physiology and development has been a subject of study for over a century[3,4,14]. The relationship between temperature and the speed of many diverse biological processes is often well approximated by the Arrhenius equation[15–22]. Originally formulated for simple, one-step chemical reactions, the Arrhenius equation describes the rate of a chemical reaction ($k$) as a function of the absolute temperature ($T$):

$$k(T) = A e^{\frac{-E_a}{RT}},\qquad(1)$$

where $E_a$ denotes a temperature-independent activation energy, $R$ is the universal gas constant, and the pre-exponential factor $A$ sets the maximal reaction rate at high temperatures. While derivable for elementary chemical reactions from thermodynamic principles, this equation is considered an empirical law applicable to various physiological rates[16]. However, deviations from this Arrhenius response are consistently observed at higher temperatures[18,22–25]. The decline is often attributed to the heat denaturation of some critical enzyme[18,25–27], and the trade-off between increasing reaction rate and increasing denaturation yields an optimal temperature. Moreover, the thermal response of processes at the cellular level is influenced not only by the reactions of enzymes but also by active cellular responses to temperature changes. For instance, in response to stress, cells may upregulate heat shock proteins, aiding in protein refolding[28]. Another example is found in budding yeast, which can up-regulate the production of viscogens like trehalose and glycogen to help maintain normal diffusion kinetics at elevated temperature[29].

Recently, the topic of biological temperature scaling has garnered renewed interest, thanks to the ability to obtain accurate, high-resolution data through time-lapse microscopy. This approach has provided fresh insights into early development in several model systems[30–33]. Overall, these findings reaffirm the utility of the Arrhenius equation as a reliable approximate description of the temperature scaling of embryonic development. This was particularly clear in studies of the timing of the first embryonic cell cycle in *C. elegans* and *C. briggsae*, two closely related nematodes. In both species, the duration of the cell cycle as a function of temperature precisely agreed with the Arrhenius equation over a broad temperature range, with some deviation then occurring when the embryos were close to their maximum tolerated temperatures[31]. The two nematodes also were found to have almost identical Arrhenius energies ($E_a$ values), which raises the possibility that the activation energies of cell cycle regulators may be evolutionarily constrained to a single standard value[31].

*Xenopus laevis* extracts and embryos have proven to be powerful systems for the quantitative analysis of the early embryonic cell cycle. The cell cycle can be experimentally studied both in vivo and in extracts, and extracts can be manipulated and observed in ways that are difficult with intact embryos. In addition, much is already known about cell cycle biochemistry in this system, providing a rich and highly quantitative context for further studies. And finally, the dynamics of the cell cycle can be successfully reproduced with relatively simple mathematical models that can add depth to the understanding of experimental findings. Crapse and colleagues have begun to examine how the *Xenopus* embryonic cell cycle is affected by temperature, and they have found some striking similarities to the behaviors seen in *C. elegans* and *C. briggsae*: the cell cycle period obeys the Arrhenius equation at least approximately, and the measured $E_a$ value for the cell cycle is similar to those in the two nematodes[32].

Here, we have leveraged the *Xenopus* system to address several outstanding questions on the principles of biological temperature scaling. First, we compared the *Xenopus laevis* temperature scaling to that in two other ectothermic vertebrate model systems, *Xenopus tropicalis* and *Danio rerio*, and compared the findings to previously reported data from the invertebrates *C. elegans*, *C. briggsae*, and *Drosophila melanogaster*. Second, we asked how well the temperature scaling is described by the Arrhenius equation, and based on ordinary differential equation modeling of the cell cycle, under what circumstances would the cell cycle be expected to exhibit Arrhenius scaling, and under what circumstances would it be expected to deviate. Finally, we asked how the different individual phases of the cell cycle vary with temperature, and found that interphase and mitosis scale differently and that this difference can be accounted for by the in vitro thermal properties of key cell cycle regulators. These studies provide insight into the principles that allow ectotherms to tolerate a range of temperatures, and suggest mechanisms for why the cell cycle oscillator fails at temperature extremes.

## Results

### Temperature scaling in the *Xenopus laevis* embryo

We measured the temperature dependence of the timing of several early cell cycle events in the developing *Xenopus laevis* embryo (Fig. 1A, B), similar to the work of Crapse and colleagues[32]. *Xenopus laevis* eggs were fertilized and imaged in a temperature-controlled chamber (first described in ref. 34) by time-lapse microscopy (Fig. 1A). We then analyzed the movies (Supplemental Movies 1, 2) to visually identify various early developmental events (Fig. 1B). First, we scored the start of the fertilization wave, a ripple in the egg's cortex that quickly spreads from the sperm entry point across the egg (at time $t_{FW}$ after fertilization). This wave is due to a trigger wave of elevated intracellular calcium, and it contributes to the block to polyspermy and coordinates the start of the cell cycle[35]. Next, we measured the start of the first surface contraction wave, which emanates from the animal pole and travels toward the vegetal pole[36,37] (at time $t_{SCW}$ after fertilization). This wave marks mitotic entry and has been argued to be caused by the interaction of a spherical wave of Cdk1 activation originating at the nucleus[38–40] with the cortical cytoskeleton[41–43]. Finally, we assessed the cleavages that complete each of the first four cell cycles. The first cleavage begins about 95 min after fertilization at 18 °C, and the next several cycles occur every 35 min thereafter[44–46]. For multicellular embryos, we took the time at which the earliest cell began to divide to be the cleavage time, but note that within an embryo, these cell divisions were nearly synchronous. The timing of all of these events was recorded for about 10 different embryos at each temperature.

Fertilized embryos reliably progressed through the cell cycle and divided at temperatures between 10 °C and 28 °C. Just outside this range (down to 9 °C and up to 29 °C) some cell cycles still occurred, and these data are included in Fig. 1. We quantified the time intervals between these developmental events and examined their temperature dependence (Fig. 1C, D), using a rearranged form of the Arrhenius equation to relate the duration of a process, $\Delta t = 1/k[T]$, to absolute temperature:

$$\ln \Delta t[T] = \ln \frac{1}{A} + \frac{E_a}{R}\frac{1}{T}.\qquad(2)$$

Accordingly, we replotted the data as $\ln \Delta t$ versus $1/T$ (Fig. 1D). Between 12 °C and 21 °C, the data were well-approximated by Eq. (2), and from the fitted slopes we extracted apparent activation energies ($E_a$) of approximately 60-80 kJ/mol (Fig. 1D and Fig. S1). An exception was the onset time of the fertilization wave, which showed a lower $E_a$ (~40 kJ/mol) with a wide confidence interval. These values are consistent with previous reports[32] and fall within the typical enzymatic range of 20–100 kJ/mol[47–49]. Outside the 12–21 °C range, the durations deviated from linearity, with unexpectedly long times at both low and high extremes (Fig. 1D). To assess the robustness of these differences,

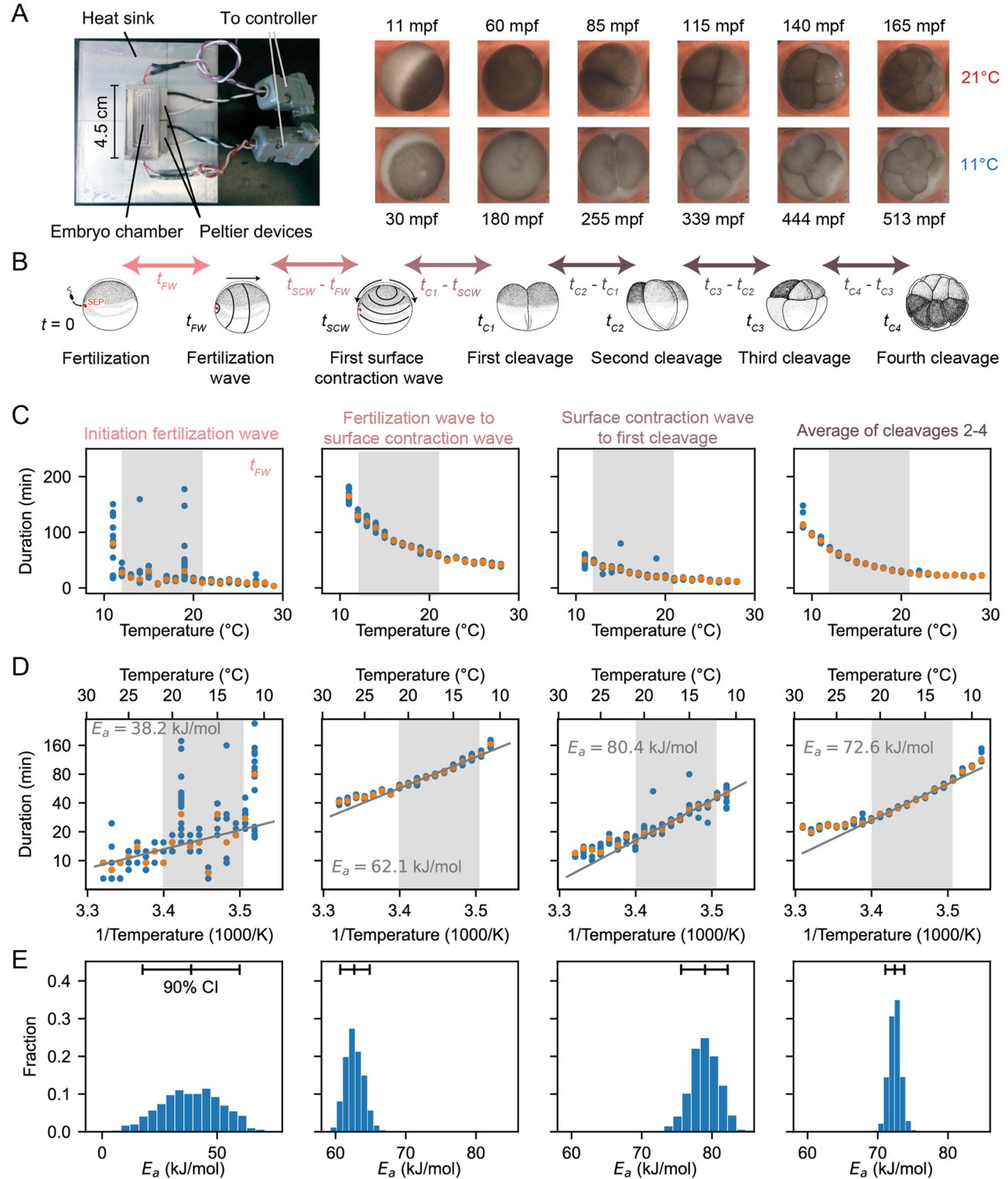

**Fig. 1 | Cell division timing in early Xenopus laevis embryos scales approximately Arrhenius over a wide range of temperatures. A** *Xenopus laevis* embryonic development was imaged in a temperature-controlled chamber introduced in ref. 34. The time unit mpf is minutes post-fertilization. **B** Different early developmental events were visually identified. SEP denotes the sperm entry point. Adapted from ref. 100. **C** Duration of several early developmental periods in function of temperature in the range [$T_{min} = 9°$ C, $T_{max} = 29°$ C]. **D** An Arrhenius fit is shown for the values between 12 °C and 21 °C, with the apparent activation energy indicated. **E** Bootstrapping provides a probability distribution for the apparent activation energies. Histograms from 1000 bootstrap samples. The mean and 90% confidence interval (CI) are also indicated. Source data are provided as a Source Data file[98].

we used a bootstrapping approach to generate probability distributions of the apparent activation energies (Fig. 1E, Fig S1, S2A and Supplementary Note 2). Finally, we computed the mean square error (MSE) between the data and the Arrhenius fit, both across the full

temperature range (9 °C to 29 °C) and within the linear range (12 °C to 21 °C). As expected, the MSE was substantially higher across the full range, confirming that the Arrhenius model does not adequately describe the entire dataset (Fig. S2B).

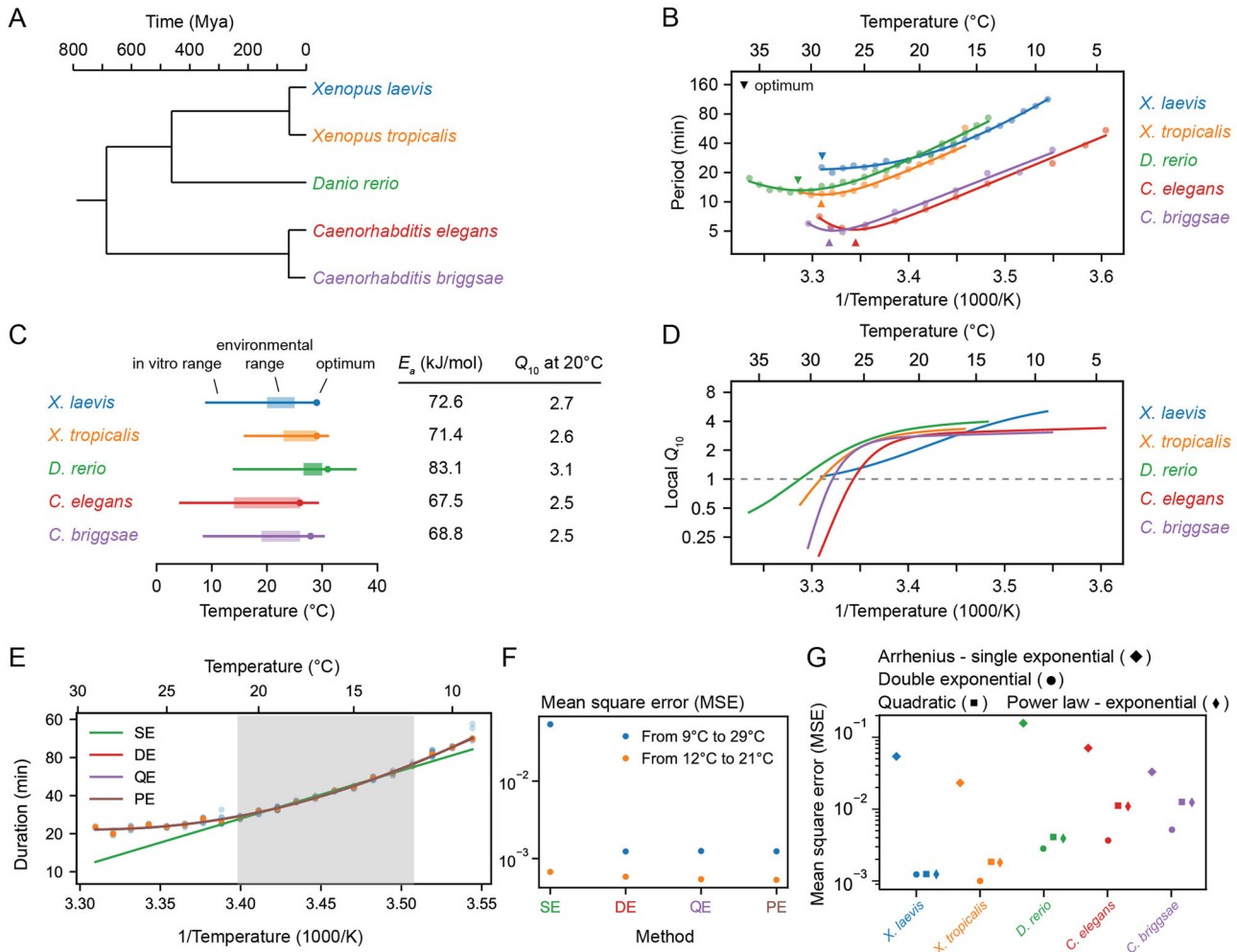

**Fig. 2 | Cell cleavage period scales in a similar non-Arrhenius way across different early ectothermic embryos. A** We examined the timing of the early cell cycles in 5 different species: *C. elegans* and *C. briggsae* (from ref. 31), *D. rerio* (this work), *X. tropicalis* and *X. laevis* (this work). The three vertebrates and the two nematodes span a broad range in evolution. **B** Median cleavage period in function of temperature for the early cell cleavages (all pooled) for the 5 different species. Optimal fits using a double exponential (DE) function are overlayed. **C** The in vivo range of viable early cell cycles in the different species, including their thermal limits and optimal temperature at which they reach a minimum cell cycle period. Their corresponding apparent activation energies and $Q_{10}$ at 20 °C are shown in the

table. Additionally the environmental range is indicated for all five organisms[101–104]. **D** Using the best DE fit, the local $Q_{10}$ value is plotted in function of temperature. **E** The median cleavage period in function of temperature for *X. laevis* is fitted using different functional forms: single exponential Arrhenius (SE), double exponential (DE), quadratic exponential (QE) and a power law-exponential (PE) function. **F** The goodness of fit (using mean square error, MSE on the logarithms of the periods) of the alternative functional forms to the experimental data for *X. laevis* in two different temperature regions: 12–21 °C and 9–29 °C. **G**. Goodness of fit, similar as in panel **F**, but now for all different species over their whole measured temperature range. Source data are provided as a Source Data file[98].

## Diverse ectothermic species yield similar temperature scaling

Next we examined the timing of the early cell cycles in two additional vertebrate model organisms, the frog *X. tropicalis* (Supplemental Movies 3, 4) and the zebrafish *D. rerio* (Supplemental Movies 5, 6). The three vertebrates and the two nematodes span a broad range in evolution (Fig. 2A).

The period of the early embryonic cell cycle as a function of temperature for all five organisms is shown in Fig. 2B. For simplicity, here we have pooled data for the durations of the early cell cleavages. In all cases, the early embryonic cell cycle could proceed over a 15–25 °C range of temperatures. In general, the five organisms showed reasonable agreement with the Arrhenius equation, especially toward the lower end of their temperature ranges (Fig. 2B). *Xenopus laevis* was something of an outlier in this regard; its Arrhenius plot is bowed throughout the temperature range (Fig. 2B, blue). The apparent Arrhenius energies—the slopes of the Arrhenius plots—were quite similar, ranging between 68 and 83 kJ/mol, or 73 ± 6 kJ/mol (mean ± std. dev., $n = 5$).

As might be expected, the nominal ambient environmental temperatures for all five organisms fell within the range found to be compatible with cell cycle oscillations (Fig. 2C). The temperature ranges for *Xenopus tropicalis* and *Danio rerio* were shifted toward higher temperatures compared to *Xenopus laevis*, reflecting the fact that the former two evolved in and live in warmer regions (Fig. 2C, orange and green vs. blue). A similar shift in the viable temperature range has been noted for the nematode worms *C. elegans* and *C. briggsae*[31], replotted here in red and purple. In all cases, the maximum temperature compatible with cycling was closer to the nominal environmental temperature range than the minimum temperature was (Fig. 2C).

For four of the five organisms (*C. elegans, C. briggsae, D. rerio,* and *X. tropicalis*) there was sufficient upward deflection of the temperature curves toward the high end of the temperature range to define an optimal growth temperature corresponding to a minimal cell cycle duration (Fig. 2C). For *Xenopus laevis,* the fastest cell cycles were found at the highest temperature compatible with viability (Fig. 2C). In all cases the optimal temperature was within a few degrees of the

maximal permissible temperature $T_{max}$ (Fig. 2C). The optimal temperatures were generally somewhat higher than the typical environmental temperature ranges (Fig. 2C). This may reflect a trade-off between maximal speed at higher temperatures and maximal safety margins in the middle of the operating temperature range. Note that at the temperature optima, the slopes of the Arrhenius plots are zero. The curves are also shallow at the optima, which means that changes of several degrees produce little changes in the cell cycle period. The period can be regarded as temperature-invariant or temperature-compensated in this regime.

Data are also available for the temperature scaling of various embryonic processes in *Drosophila melanogaster*. Extensive data are available for the timing between the 13th and 14th cleavage[32,50], and these data are replotted in Fig. S3A, B. Like the *Xenopus laevis* Arrhenius plot, the *Drosophila* plot is bowed throughout the temperature range, and, at the cold end of the temperature range, it yielded an Arrhenius energy of 109 kJ/mol. Note, however, that this cycle differs from the earlier *Drosophila* cycles and the other embryonic cycles examined here, as they have a longer cell cycle due to lengthening of S-phase approaching the mid-blastula transition (MBT)[50,51]. Some data are available for the 11th nuclear cycle (NC11) in the syncytial *Drosophila* embryo (Fig. S3A)[33]; this cycle is more similar to the other organisms' cycles analyzed here. From the published data, we calculated an Arrhenius energy of 84 kJ/mol for the duration of NC11, slightly higher than the energies calculated for the early embryonic cycles of the other 5 model organisms. Taken together, the six organisms yielded an average Arrhenius energy for the early embryonic cell cycles of $75 \pm 7$ kJ/mol (mean ± std. dev.). Although the nominal periods of the cell cycles varied greatly, from about 5 min for *C. elegans* and *C. briggsae* to 25 min for *X. laevis* at room temperature, the temperature scaling factors for these organisms varied by only about 9%.

The temperature sensitivity of the early embryonic cell cycle can also be characterized by $Q_{10}$ values, which capture how reaction rates change over 10 °C intervals[52]. While linear Arrhenius fits give a global $Q_{10}$ value around 2.8 across diverse organisms (Fig. 2C), local $Q_{10}$ values (see Supplementary Note 1) reveal important deviations, particularly at temperature extremes (Fig. 2D). For most species, local $Q_{10}$ values plateau around $2.8 \pm 0.4$ at low temperatures and decrease at higher temperatures. However, in *Xenopus laevis*, local $Q_{10}$ values vary more broadly, ranging from about 1 to 4 across the full temperature range (Fig. 2D and Fig. S3E).

To better capture the deviations from idealized Arrhenius behavior, we explored three commonly used generalizations of the Arrhenius equation-the double exponential[31,53,54], quadratic exponential[32,55], and power law-exponential forms[56,57] (see "Methods" for details on the fitting). All three models provided markedly better fits to the experimental data than the classical Arrhenius relationship, capturing the nonlinearities in the temperature dependence with high accuracy (Fig. 2E–G). However, their comparable performance makes it difficult to identify a single best functional form based on fitting alone. This underscores the limitations of descriptive models and points to the need for mechanistic frameworks to explain the origin of temperature scaling curves in biological systems.

## A simple oscillator model accounts for deviations from Arrhenius scaling

We next took a computational approach to the question of why cell cycle periods at least approximately obey the Arrhenius equation, and why they sometimes deviate from Arrhenius scaling. We used a differential equation model of the embryonic cell cycle oscillator to investigate how the period would be expected to vary with temperature, given either single or double exponential equations for the individual enzymes' temperature scaling. This allowed us to examine how systems-level properties of the cell cycle oscillator circuit, rather than just variations from the Arrhenius relationship in the behaviors of

individual enzymes, might be expected to affect the temperature scaling of the oscillations.

The cell cycle regulatory network consists of many complex interactions involving dozens of species, which makes it extremely challenging to construct a complete mathematical model, let alone study and interpret the influence of temperature on the cell cycle. The early embryonic cell cycle of insects, worms, amphibians, and fish is, however, much simpler: the cycle consists of a rapidly alternating sequence of synthesis (S) phase and mitotic (M) phase, without checkpoints and without G1 and G2 gap phases. Transcription is negligible at this point in embryogenesis, and the number of protein species involved is smaller than in the somatic cell cycle. As a result, simpler mathematical models can be constructed. This greatly simplifies the analysis of temperature scaling.

At the heart of the early embryonic cell cycle lies the protein complex cyclin B · Cdk1, consisting of the protein cyclin B and the cyclin-dependent-kinase Cdk1 (Fig. 3A), plus a phospho-epitope-binding subunit, Suc1/Cks, that is not separately considered here. When this protein kinase complex is enzymatically active, it phosphorylates hundreds of other proteins, bringing about the entry of the cell into mitosis[58,59]. The oscillations in Cdk1 activity result from three interlinked processes: (1) cyclin B synthesis, which dominates in interphase and causes Cdk1 activity to gradually rise; (2) the flipping of a bistable switch, due in this model to the Cdk1-Cdc25 positive feedback loop and the Cdk1-Wee1 double negative feedback loop, which results in an abrupt rise in Cdk1 activity; and (3) cyclin B degradation by the anaphase-promoting complex/cyclosome APC/C and the proteasome, which dominates during M-phase and causes Cdk1 activity to fall (Fig. 3B)[60–62]. This type of oscillator, consisting of a rapid bistable switch plus a negative feedback loop, is referred to as a relaxation oscillator[63–65]. Relaxation oscillators are common in biology, and all relaxation oscillators share similar qualitative behavior, irrespective of the exact molecular details: there is a slow ramp up in activity, which then triggers an abrupt burst in activity through the positive feedback loop(s), and finally, the negative feedback restores the system back to its low activity state. These three distinct phases can be distinguished in experimental data on the activity of Cdk1 as a function of time in the early embryonic cell cycle (Fig. 3C; see also below).

We described the changes in cyclin concentration and Cdk1 activity with a model consisting of two ordinary differential equations (ODEs)[66]:

$$\frac{d\text{cyc}[t]}{dt} = k_s - k_d \text{APC}_a[t] \, \text{cyc}[t],$$
$$\epsilon \frac{d\text{cdk1}_a[t]}{dt} = k_a \text{Cdc25}_a[t]( \text{cyc}[t] - \text{cdk1}_a[t]) - k_i \text{Wee1}_a[t]\text{cdk1}_a[t]. \quad (3)$$

The first equation describes how cyclin B (cyc) is synthesized throughout the cell cycle at a rate $k_s$ (nM/min) and how it is degraded at a rate $k_d$ (1/min) by the proteasome after ubiquitination by active APC/C (APC$_a[t]$). The second equation describes the conversion of cyclin B-Cdk1 complexes between an inactive form and an active form by Cdc25 (Cdc25$_a[t]$) and Wee1 (Wee1$_a[t]$). For simplicity, we do not directly include degradation of the bound-form of cyclin B (see Supplementary Note 3.A). If we assume that the Cdk1-mediated phosphorylation reactions that regulate APC/C, Wee1, and Cdc25 are essentially instantaneous, we can eliminate three of the time-dependent variables from the right-hand side of the ODEs:

$$\frac{d\text{cyc}[t]}{dt} = k_s - k_d d[\text{cdk1}_a] \, \text{cyc}[t],$$
$$\epsilon \frac{d\text{cdk1}_a[t]}{dt} = k_a a[\text{cdk1}_a]( \text{cyc}[t] - \text{cdk1}_a[t]) - k_i i[\text{cdk1}_a]\text{cdk1}_a[t]. \quad (4)$$

The terms $d[\text{cdk1}_a]$, $a[\text{cdk1}_a]$, and $i[\text{cdk1}_a]$ are assumed to be Hill functions of the instantaneous values of cdk1$_a$, and were

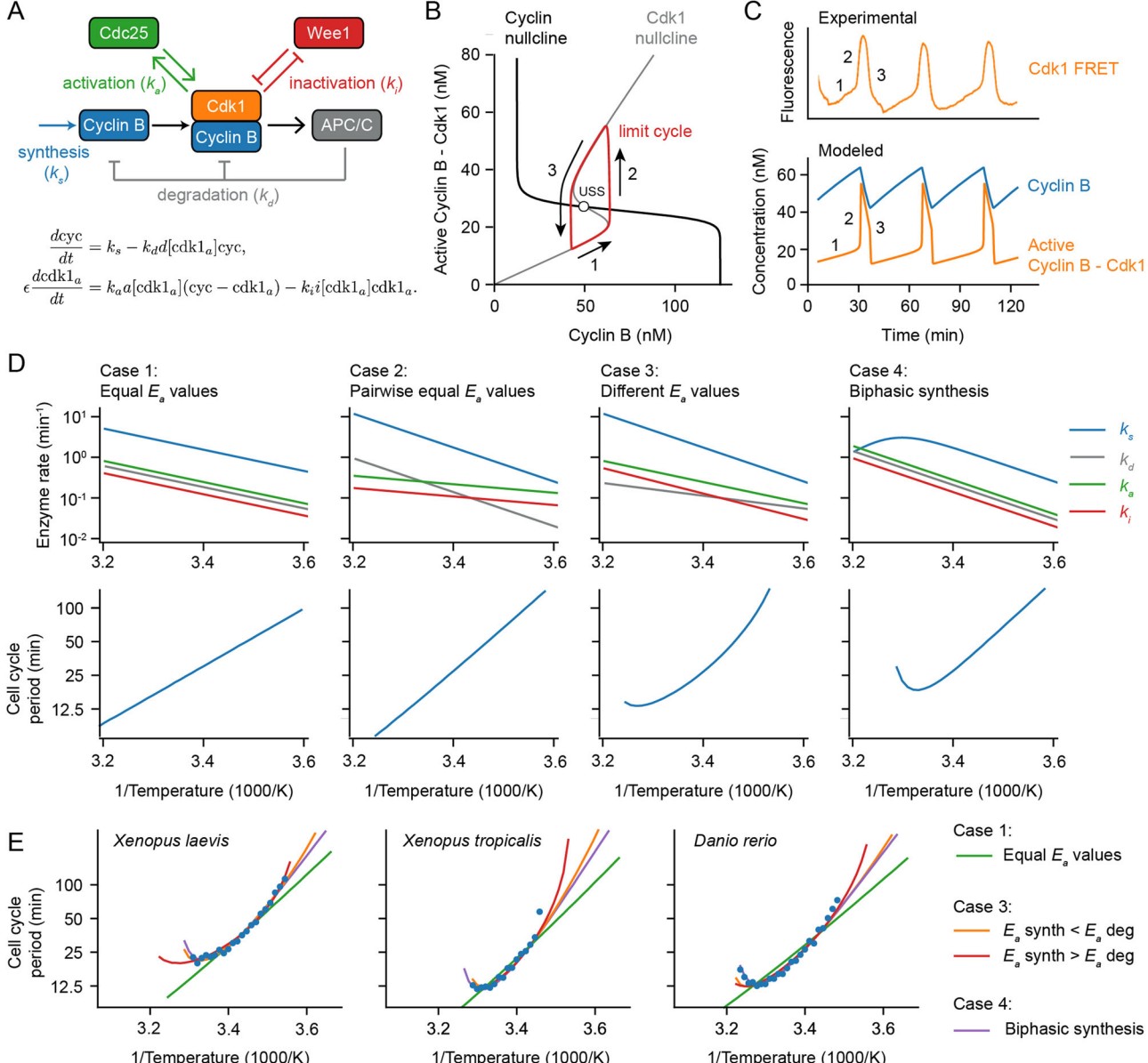

**Fig. 3 | A simple relaxation oscillator model for the early embryonic cell cycle can reproduce the observed non-Arrhenius scaling. A** Sketch of key reactions in the early cell cycle regulatory network. **B**, **C** Phase plane representation (**B**) and time series (**C**) of cell cycle oscillations in Eq. 4. **D** Scenarios showing how different temperature scaling of cell cycle regulatory processes can lead to Arrhenius scaling and/or thermal limits in the scaling of the cell cycle period. **E** Best fits of models presented in panel (**D**) to the measured data for the early cell cycle duration for *X. laevis*, *X. tropicalis* and *D. rerio* shown in Fig. 2. For case 3, the apparent activation energies for $k_s$ and $k_d$ need to be different to fit the data well. For Case 4, we introduced a biphasic response in cyclin B synthesis ($k_s$). For parameter values and more details about the model, see Supplementary Note 3. Source data are provided as a Source Data file[96].

parameterized based on experimental measurements of steady-state responses in *Xenopus laevis* extracts[66–68]. We thus have two ODEs in two time-dependent variables, cyc[$t$] and cdk1$_a$[$t$], and five parameters that define the speeds of cyclin synthesis ($k_s$), cyclin degradation ($k_d$), Cdk1 activation by Cdc25 ($k_a$), and Cdk1 inactivation by Wee1 ($k_i$), as well as the relative time scales of cyclin synthesis and degradation versus Cdk1 activation and inactivation ($\epsilon$). Even though this simplified model omits Greatwall/PP2A-B55 regulation and a number of other interesting aspects of *Xenopus* cell cycle regulation[62,65,69–72], it nevertheless captures the dynamics of Cdk1 activation and inactivation well and has been used to successfully describe various aspects of cell cycle oscillations[34,66,73]. For this reason, it seemed like a good starting point for understanding how the output of the cell cycle oscillator circuit would be expected to scale with temperature.

Using experimentally motivated parameters[66], the model (Eq. (4)) reproduced cell cycle oscillations with a realistic period of approximately 30 min (Fig. 3C). These oscillations manifested as a closed trajectory, a limit cycle, in the (cyc, cdk1$_a$) phase plane (Fig. 3B, red), which orbits around an unstable steady state (Fig. 3B, USS) at the intersection of the system's two nullclines (Fig. 3B). When the time scale for Cdk1 activation/inactivation is fast relative to the time scale of cyclin synthesis and degradation ($\epsilon \ll 1$), typical relaxation oscillations occur: in interphase, the orbit slowly creeps up the low Cdk1 activity portion of the S-shaped nullcline (Fig. 3B, denoted 1), then abruptly jumps up to the high Cdk1 activity portion of the same nullcline (Fig. 3B, denoted 2), crawls down the nullcline due to active APC/C (Fig. 3B, denoted 3), and abruptly falls back down to the lower portion of the nullcline to begin the cycle again. The result is sawtooth-shaped

oscillations in cyclin B levels and periodic bursts of Cdk1 activity that resemble experimentally measured Cdk1 activities (Fig. 3C).

Next we examined how making the model's parameters temperature-dependent affected the persistence and period of the oscillations over a range of temperatures. We started by assuming Arrhenius scaling for the four key rates in the oscillator model: $k_s$, $k_d$, $k_a$, and $k_i$. As expected, when all of the apparent activation energies were assumed to be equal, the oscillation period obeyed the Arrhenius equation with the same activation energy (Case 1 in Fig. 3D). Similarly, if the activation energies for cyclin synthesis and degradation were assumed to be equal, and the energies for Cdk1 activation and inactivation were assumed to be different from those but equal to each other, the period also scaled in an Arrhenius fashion (Case 2 in Fig. 3D), with $E_a$ equal to that of cyclin synthesis and degradation. This Arrhenius scaling arises out of three properties of the model: (1) the cyclin nullcline's position depends solely on the ratio $k_s/k_d$, and these two parameters were assumed to scale identically with temperature; (2) the location of the S-shaped Cdk1 nullcline depends upon the ratio $k_a/k_i$, which likewise was assumed to scale identically with temperature; and (3) as long as $\epsilon$ is very small, the cell cycle period is determined only by the rates of cyclin synthesis and degradation, which scale identically with temperature.

However, if the four kinetic parameters were not constrained to scale identically (Case 1) or pair-wise identically (Case 2) with temperature, the results were more like what is seen experimentally. This is shown as Fig. 3D, Case 3. The Arrhenius plot was bowed concave up, instead of being straight, and oscillations ceased if the temperature was too high or too low. The cessation of oscillations can be rationalized from the positions of the nullclines in the phase plane. If the $E_a$ value for cyclin synthesis is smaller than that for cyclin degradation, the ratio $k_s/k_d$ decreases as temperature rises. With increasing temperature, the cyclin nullcline shifts down until it no longer intersects the middle portion of the S-shaped Cdk1 nullcline. At this point, the steady state becomes stable and oscillations cease, leaving the system in an interphase-like steady state with low Cdk1 activity (Fig. S4A). Conversely, if cyclin synthesis scales more strongly with temperature than degradation does, the cyclin nullcline shifts upward, leading to a stable M-phase-like steady state with high Cdk1 activity (Fig. S4C). Thus, if the temperature scaling of cyclin synthesis and degradation differ, at extremes of the temperature range, the oscillator will fail, and in between the extremes, the cell cycle period would be expected to deviate from the Arrhenius relationship. This could provide an explanation for the temperature scaling observed experimentally (Fig. 1).

An alternative assumption could also explain the experimental results. As shown in Fig. 3D, Case 4, if at least one process shows a biphasic dependence of rate on temperature, perhaps due to enzyme denaturation at high temperatures, the result will be a bowed Arrhenius plot and a high temperature limit to oscillations. As an example, here we have assumed a biphasic dependence of cyclin synthesis on temperature. Thus, in principle, it seemed like either variation in the individual enzymes' Arrhenius energies (Case 3), or denaturation at high temperatures (Case 4), or both, could account for the observed temperature scaling of the early embryonic cell cycle.

To test this hypothesis further, we asked how well model Cases 3 and 4 could replicate the observed cell cycle duration scaling in early frog and zebrafish embryos. We adjusted the apparent activation energies of cyclin synthesis and degradation, and, for simplicity, kept the $E_a$ values for Cdk1 activation and inactivation constant. Through minimizing the error between simulations and data across the oscillation range, employing the mean sum of squares on logarithms of periods, we obtained optimal fits. Figure 3E displays these fits across various model scenarios introduced in Fig. 3D. Although perfect Arrhenius scaling (Cases 1 and 2) did not align well with the data, model Cases 3 and 4 approximated the measured data well. Notably, the experimental data could be accounted for by assuming that cyclin

synthesis scaled either more strongly or more weakly with temperature than did cyclin degradation.

To further reinforce these findings, we conducted an exhaustive parameter scan over the activation energies of all four key rates in the oscillator model using a fitting algorithm. We employed the approximate Bayesian computation method[74], implemented in Python using pyABC[75], for sequential Monte Carlo sampling of parameter sets, gradually improving fits to the data (for details, see Supplementary Note 3.D and Fig. S5). This broader analysis underscored that optimal fits occurred when there was a distinct difference in apparent activation energies between cyclin synthesis and degradation, while the activation energies of Cdk1 activation and inactivation remained similar (Fig. S5). Moreover, the quantification of fitting errors revealed that it is most probable that the experimental data is due to cyclin synthesis being more sensitive to temperature changes than cyclin degradation ($E_a(k_s) > E_a(k_d)$).

Next, we asked whether the results obtained were specific to the two-ODE cell cycle model. To explore this, we turned to a structurally distinct model: a five-ODE, mass-action-based system that includes interactions among Cdk1, Greatwall, and PP2A-elements that collectively form a mitotic switch as well[62,65,76]. In contrast to the two-ODE model, which featured a bistable switch between Cdk1 and Cyclin B and included feedback through Cdc25 and Wee1, this model implements a bistable switch between APC/C and Cdk1, and thus omits the Cdc25/Wee1-mediated feedback loops entirely. It also differs in its use of strictly mass-action kinetics, avoiding the highly nonlinear Hill functions of the two-ODE model, and in its dimensionality, expanding from two to five ODEs. Despite these structural differences, both models share key features: cyclin synthesis and Cdk1-activated degradation, the presence of a bistable switch, and a separation of timescales enabling relaxation oscillations. The five-ODE model could be parameterized to yield realistic cell cycle oscillations[77] (Fig. S6B and Supplementary Note 3.B). Due to the model's increased complexity— ten kinetic parameters—we relied exclusively on the ABC algorithm for parameter inference. This approach produced satisfactory fits (Fig. S6E), and analysis revealed that highly correlated activation energy pairs typically corresponded to antagonistic reaction rates (Fig. S6F, G and Supplementary Note 3.B). These results again show that well-fitting parameter sets tend to exhibit similar activation energies for faster reactions. Moreover, they support the idea that thermal limits can arise from imbalances in the apparent activation energies of cyclin synthesis and degradation, reinforcing the conclusions drawn from the two-ODE model.

In summary, computational modeling revealed that thermal limits and non-Arrhenius scaling like those seen in early embryos can arise from (at least) two different mechanisms. Firstly, in cases where all rates follow Arrhenius-like scaling but possess varying activation energies, an imbalance emerges, culminating in a thermal limit and a bowed Arrhenius plot. We can call this behavior 'emergent', since the limit and the bowing are not inherent to any individual reaction but arise collectively. Secondly, thermal limits can arise if one or more underlying reactions exhibit a thermal optimum and deviate from Arrhenius scaling. Here, the system's behavior is predominantly dictated by the dynamics of the particular biphasic component(s).

### The durations of interphase and M-phase scale differently with temperature

To test whether the emergent imbalance model (Fig. 3C, Case 3) contributes to the temperature scaling of the *Xenopus laevis* embryo, we set out to determine how the durations of interphase and M-phase individually scaled with temperature. Both phases contribute to the overall duration of the cell cycle, and the durations of the two phases are largely determined by different processes, cyclin synthesis for the former and cyclin degradation for the latter. Due to the opacity of the *Xenopus* embryo, it is difficult to assess these cell cycle phases by

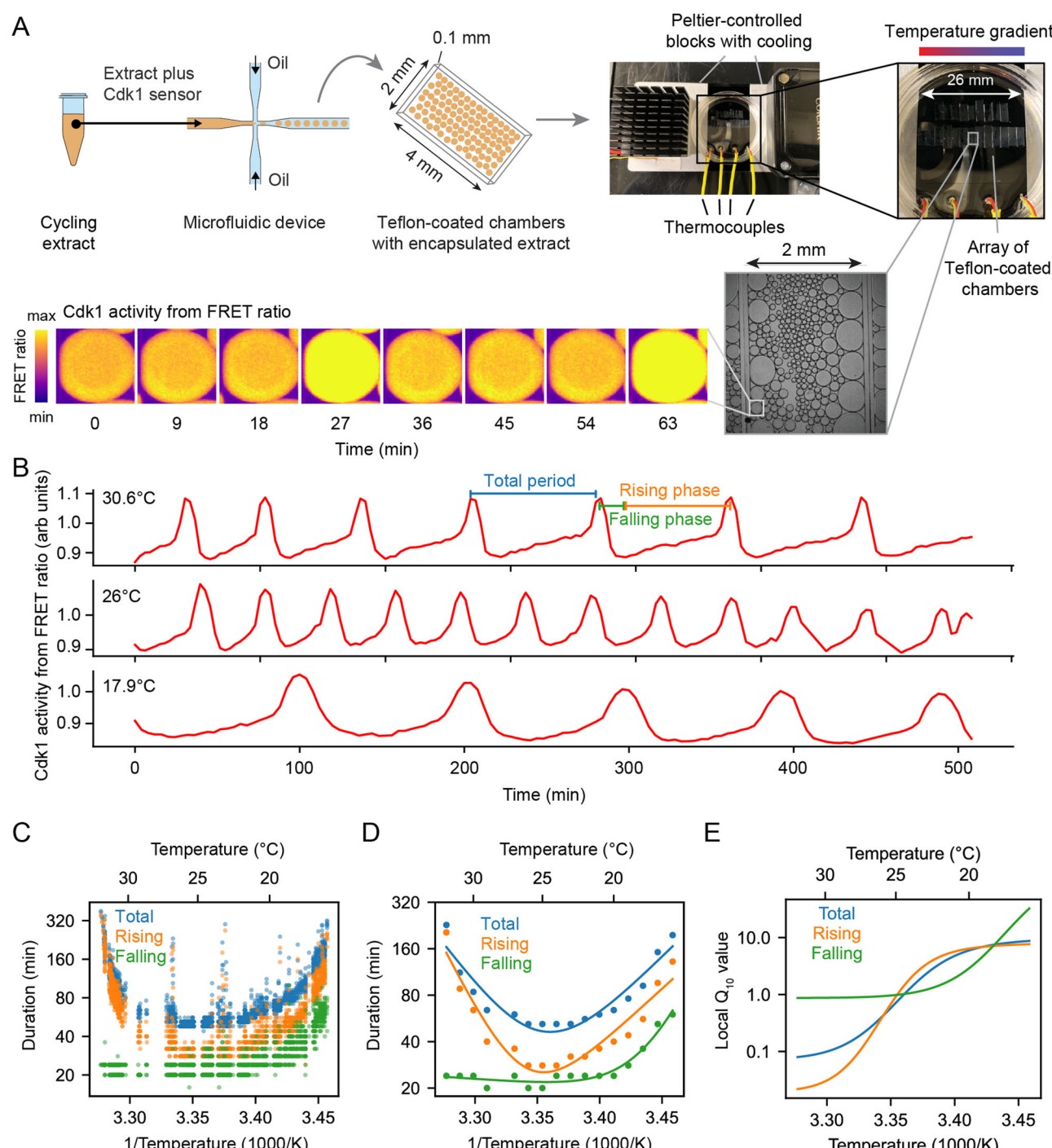

**Fig. 4 | The durations of interphase and M-phase scale differently and non-Arrhenius with temperature in cycling frog egg extracts. A** Sketch of the setup to encapsulate cycling frog egg extracts in droplets surrounded by oil, including pictures of the customized device to control temperature of extract droplets with snapshots of measured FRET ratios in an example droplet. **B** Representative time series of measured FRET ratios at different temperatures. **C** Analysis of the duration of the total cell cycle (blue), the rising phase (orange), and the falling phase (green) in function of temperature. **D** Median per temperature bin (rounded to integers) of the data shown in panel (**C**). Optimal fits using a double exponential function are overlaid. **E** Local $Q_{10}$-value as a function of temperature, calculated from the fitted double exponential function. Source data are provided as a Source Data file[96,98].

in vivo microscopy. We therefore turned to cycling *Xenopus* egg extracts, which are transparent and highly amenable to microscopy.

Cycling extracts were prepared and supplemented with a Cdk1 FRET sensor, whose emission increases upon Cdk1 activation and/or inactivation of opposing phosphatase(s)[78]. Unlike the original PBD (polo-box domain)-based sensor used in human cells[79] and *Drosophila* embryos[80], this redesigned WW-based sensor uses a Cdc25C substrate motif, WW phospho-binding domain, and EV linker to enhance signal-to-

noise performance in *Xenopus* extracts[78]. The extracts were then encapsulated in oil droplets, as previously described[81,82] (Fig. 4A). The encapsulated extract droplets were then loaded into Teflon-coated imaging chambers, which were immersed in mineral oil and placed on a microscope stage equipped with a custom Peltier element-based heating/cooling device, similar to a setup tailored for embryos[44]. The FRET sensor enabled real-time visualization of oscillations in Cdk1 activity in hundreds of droplets situated at different positions within the

temperature gradient (Fig. 4A and Supplemental Movie 7). As shown in Fig. 3C (top), Cdk1 activity first rose slowly (phase 1), then spiked to high levels (phase 2), then fell to low levels to allow a new cycle to begin (phase 3). The three phases of the Cdk1 activity cycle correspond well to the phases seen by direct biochemical assays of Cdk1 activities in cycling extracts[62,83,84], and to the phases of Cdk1 activation and inactivation seen in the computational models (Fig. 3C, bottom). The cell cycle was found to proceed most rapidly at temperatures of around 25 °C, and to slow down at both colder and warmer temperatures (Fig. 4B, C). One unanticipated finding was that the cell cycle proceeded fairly normally at temperatures as high as 32 °C, even though in intact embryos, temperatures above 28 °C typically killed the embryos and halted the cell cycle. This allowed us to probe a wider range of temperatures in extracts than was possible in vivo. The period of the extracts' cell cycles increased over time, consistent with previous findings[39,78,81,85]. In addition, the overall response of the cell cycle to temperature was consistent across different biological samples and experimental days (Fig. S7). We confined our analysis to cycles 2–4, characterizing the duration of each cycle in individual droplets (Fig. S8A–C) and after pooling (Fig. 4C, D). Similar trends could be seen in both the individual and pooled data (Fig. 4C, D). Alternatively, we analyzed all cell cycles that occurred during the first 300 min rather than the first three cycles. This procedure yielded essentially identical results (Fig. S8D).

We analyzed time series data from hundreds of droplets at temperatures from 16 °C to 32 °C, and plotted the temperature dependence of the cell cycle durations as well as the durations of the rising and falling phases, which correspond approximately to interphase-through-metaphase and metaphase-through-mitotic exit, respectively. This analysis showed that both the total cell cycle duration and the duration of the rising phase exhibited a U-shaped dependence upon temperature. These durations decreased steeply as the temperature rose from 16 °C to 20 °C, then plateaued, and then increased steeply at temperatures above 30 °C (Fig. 4C, D). In contrast, the duration of the falling phase decreased with temperature and then plateaued beginning at about 20 °C, but did not slow down to a measurable extent at higher temperatures. These trends can be seen both from the raw data (Fig. 4C) and from binned, averaged data (Fig. 4D). Thus, the rising phase, whose duration is mainly due to the rate of cyclin synthesis, and the falling phase, whose duration is mainly due to APC/C activity, are differently affected by temperature. Interestingly, while the durations of transitioning into and out of M phase increased at low temperatures, the duration of mitotic exit remained constant at high temperature (Fig. S9). Conversely, at high temperatures, the duration of mitotic entry substantially increased.

Double exponential curves, which assume a biphasic dependence of enzyme activity upon temperature, accounted for the shapes of the Arrhenius plots (Fig. 4D). We computed local $Q_{10}$ values from the fitted curves, which revealed significant changes with temperature (Fig. 4E). Generally, the $Q_{10}$ for total cell cycle duration was close to that of rising phase duration (Fig. 4E), underscoring the fact that interphase constitutes a majority of the cell cycle (Fig. 4B).

## Cyclin synthesis and degradation respond differently to temperature

We next asked how well the 2-ODE computational model could account for how the Cdk1 activity cycle varied with temperature in cycling extracts, and whether the scaling of the activation energies for key regulatory processes ($k_s$, $k_d$, $\epsilon$) could be inferred. We utilized the temperature dependence of the measured durations of the rising and falling phases of the Cdk1 time series (Fig. 4D) to optimize our computational model. Employing the approximate Bayesian computation method (details in Supplementary Note 3.D and Figs. S10, S11), we sought optimal values that described the scaling curves for cyclin synthesis rate ($k_s$), cyclin degradation rate ($k_d$), and time scale separation ($\epsilon$), which relates to Cdk1 activation ($a$) and inactivation ($i$). The temperature dependence

of each of these parameters is described by a double-exponential scaling curve (for details, see Supplementary Note 3.D). Leveraging sequential Monte Carlo sampling, the method gradually improved fits to the data. Rather than a single optimal value, the method produces a distribution of well-fitting parameters (gray lines in Fig. 5A, with the optimal fit highlighted in color). The temperature dependence of the fitted model parameters (Fig. 5B) revealed significant temperature-induced changes in cyclin synthesis rate ($k_s$) and time scale separation ($\epsilon$), up to five-fold across the temperature range, whereas changes in the cyclin degradation rate ($k_d$) were much smaller (see Figs. S10, S11 for the distributions of the parameters yielding good fits). Additionally, the temperature dependence of $k_d$ was well described by a single Arrhenius equation, while $k_s$ and $\epsilon$ required a double exponential function for accurate description. Figs. S10, S11 further demonstrate that the model successfully captures the observed changes in oscillation dynamics as long as the cyclin synthesis rate is more temperature-sensitive than the degradation rate. This means that the fitted values of these activation energies are not tightly constrained by the data. Specifically, the model remains consistent with the data as long as the cyclin synthesis rate increases more steeply with temperature than the degradation rate. This is supported by simulation results and by the ABC-inferred parameter distributions (Figs. S10, S11), which show that the apparent activation energy for synthesis, $E_a(k_s)$, is centered around 113 kJ/mol. In contrast, $E_a(k_d)$ for degradation is very broadly distributed, and although the peak is at around 12 kJ/mol, the mean is closer to 49 kJ/mol. This suggests that a range of degradation temperature sensitivities is compatible with the data, provided synthesis remains more sensitive.

The three phases of the Cdk1 activity cycle correspond well to the phases seen by direct biochemical assays[61,62,84], and they represent the accumulation of low activity cyclin-Cdk1 complexes (phase 1), followed by the activation of Cdc25 and the inactivation of Wee1 and PP2A-B55 (phase 2), and finally the APC/C-Cdc20-mediated degradation of cyclin and re-activation of PP2A-B55 (phase 3). During the first phase, both cyclin levels and Cdk1 activity increase approximately linearly over time[61,62] (Fig. 5B, C). To obtain an independent estimate of how $k_s$ varies with temperature, we computed the slope of the interphase segment of the Cdk1 FRET time series (Fig. 5C, dashed line; Supplementary Note 4, Fig. S12). These empirical slopes revealed scaling trends consistent with those obtained from model fitting via the ABC algorithm (Fig. 5B, overlaid black dots), reinforcing the robustness of the inferred biphasic temperature dependence. While the Cdk1 FRET signal primarily reflects cyclin accumulation, it could in principle also be influenced by other regulatory processes, such as phosphatase inactivation or post-translational modifications that modulate Cdk1 activity independently of cyclin levels. Therefore, fitting the slope of the FRET signal does not necessarily isolate cyclin synthesis alone. Nonetheless, the strong agreement between the model's predictions and the slope-based estimates suggests that the FRET signal serves as a useful proxy for cyclin synthesis over this regime. This interpretation is further supported by previous studies showing that cyclin synthesis is the primary driver of Cdk1 activation during interphase. Pomerening et al. demonstrated that both cyclin levels and Cdk1 activity rise approximately linearly with time throughout interphase[61], consistent with a model in which increasing concentrations of phosphorylated, pre-activated cyclin-Cdk1 complexes underlie the gradual activation of the oscillator. Moreover, other key regulators of Cdk1, including Cdc25C and Wee1, exhibit minimal changes in their abundance or phosphorylation state during interphase and only transition into their mitotic forms immediately before mitotic entry (see Fig. 7 in[62]).

As a final check of our fitting procedure, we computed the average shape of the Cdk1 time series experimentally (from the FRET signal, Fig. S13, Supplementary Note 5), and compared it to the model's predicted time series, given the fitted scaling (Fig. 5B). The simulated time series closely recapitulated the experimental oscillations (Fig. 5C). The optimization was done only on durations. Thus, the match between

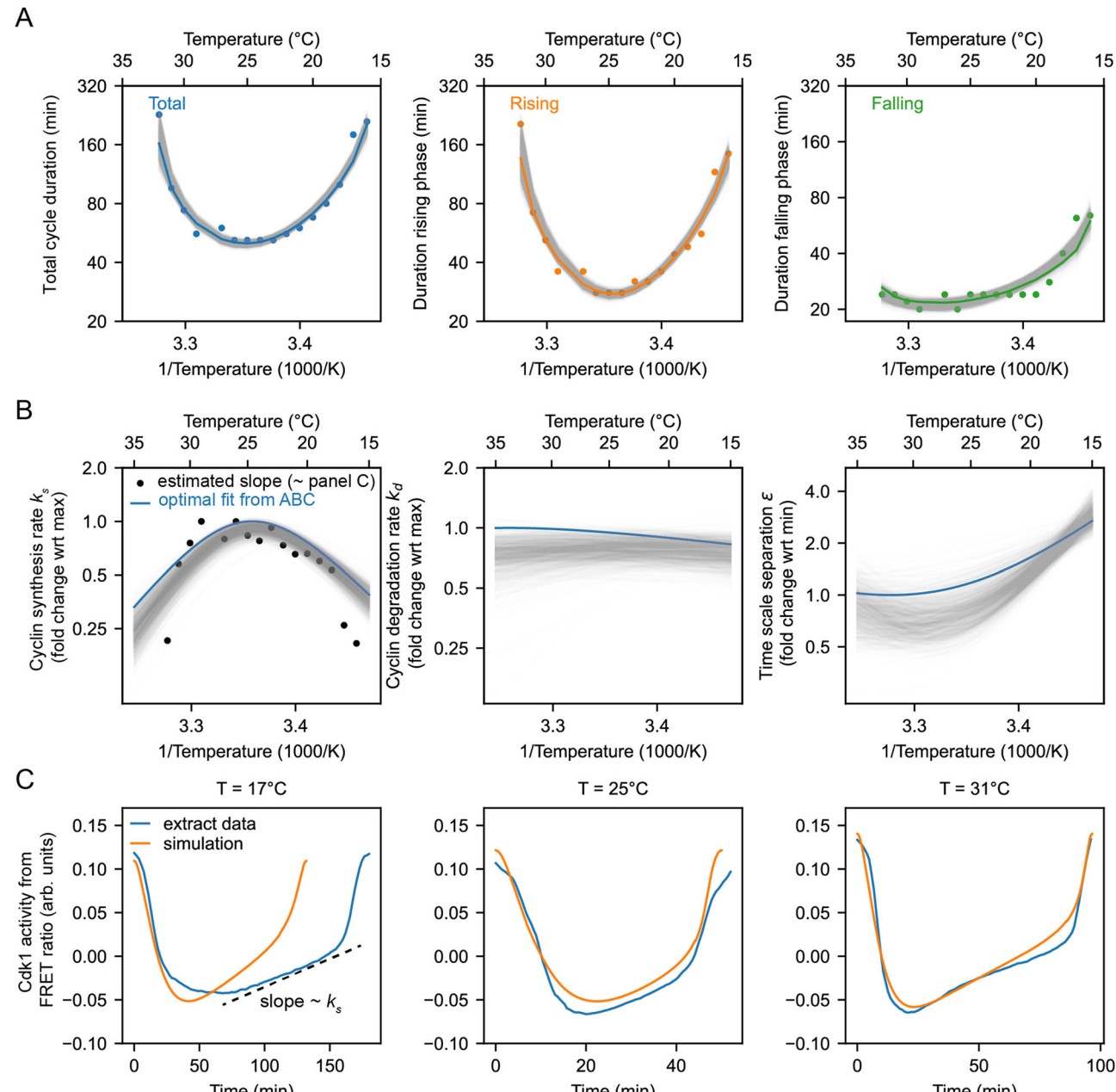

**Fig. 5 | Non-Arrhenius scaling as a result of biphasic cyclin synthesis rate and an imbalance in the temperature scaling of cyclin synthesis and degradation.**
**A** Using the ABC algorithm, we minimize the mean square error (MSE) between the measured and simulated (using the two-ODE model) durations of rising phase and falling phase. The measurements are shown with the dots. Each gray line shows the result of one parameter set from the outcome of the ABC algorithm (darker gray means larger weight). The colored line shows the best fit (curve resulting from parameter set with the smallest MSE). **B** Optimal temperature scaling of parameters, i.e., the cyclin synthesis rate, the cyclin degradation rate, and the time scale separation, resulting from the ABC algorithm as shown in (**A**). The black dots correspond to the cyclin synthesis rate $k_s$ (nM/min) directly estimated from the FRET ratio time series (Fig. S12 and Supplementary Note 4). **C** Blue line: averaged time series of Cdk1 activity (measured FRET ratio) at three different temperatures ($T = 17\,°C$, $T = 25\,°C$, $T = 31\,°C$). See Fig. S13 and Supplementary Note 5 for the method to compute the average waveform. Orange line: time series of the computational model, computed using the optimal parameter scaling shown in Panel **B**. Source data are provided as a Source Data file[96,98].

simulated and experimentally observed waveforms provides another argument that the fitted scaling curves for the rates explain the scaling observed in the droplets.

In summary, the comparison of apparent activation energies highlights the greater temperature sensitivity of cyclin synthesis rate compared to cyclin degradation rate. This sensitivity aligns with scenarios predicted to yield non-Arrhenius scaling across a wide temperature range (Fig. 3D, Case 3). Furthermore, experimental findings indicate that cyclin synthesis rates decreased at elevated

temperatures, corroborating another scenario leading to non-Arrhenius scaling (Fig. 3D, Case 4). Our analysis indicates that both mechanisms contribute to the non-Arrhenius scaling properties of the early embryonic cell cycle oscillator.

**In vitro assays confirm the imbalance in cyclin synthesis and degradation scaling**
To further test the inference that cyclin synthesis and degradation scale differently at the low end of the temperature range, we carried out direct

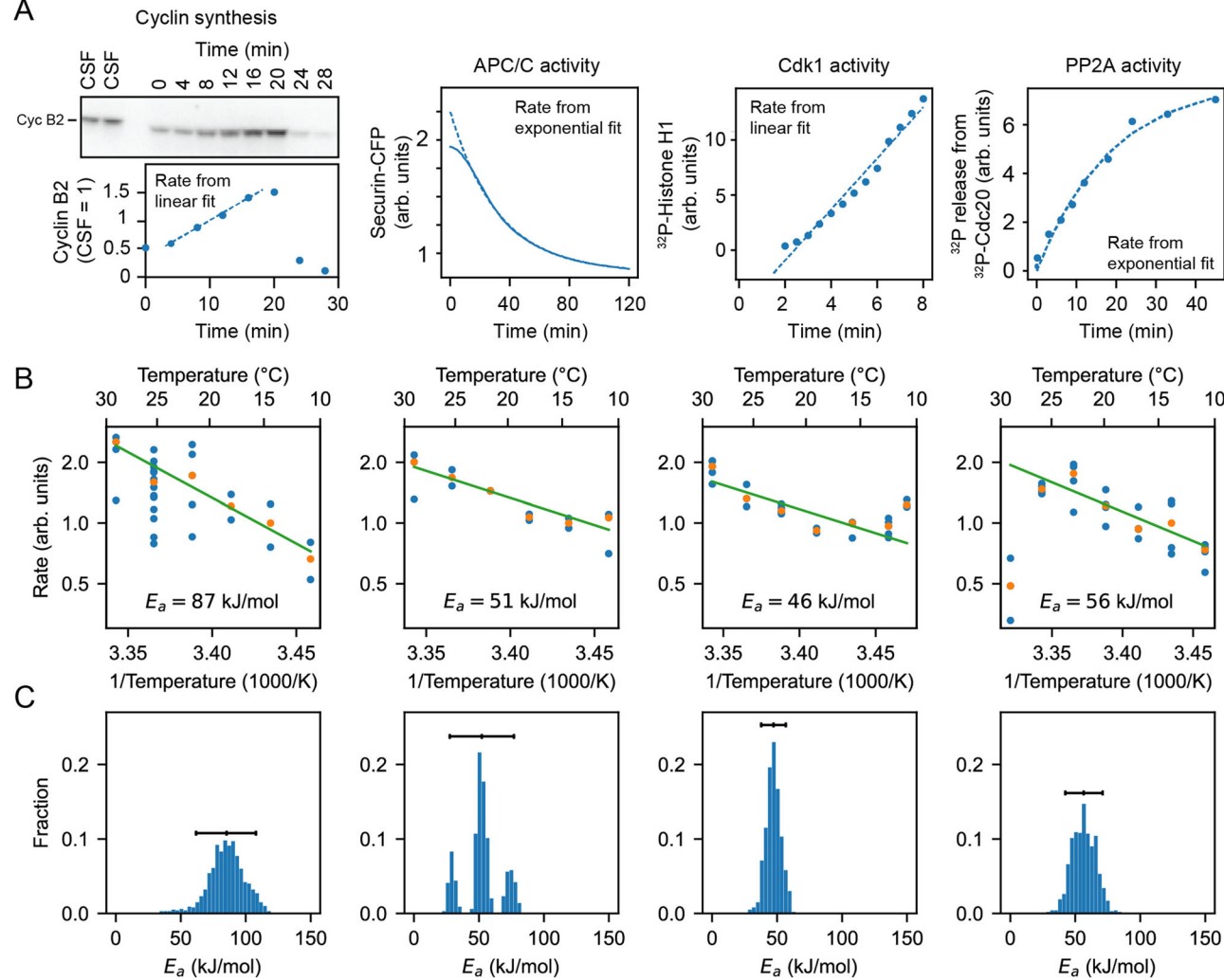

**Fig. 6 | Frog egg extract measurements reveal temperature dependence of cell cycle regulators. A** Examples for how the rates for Cyclin B synthesis, APC/C activity, Cdk1 activity, and PP2A activity were fitted from time series of different biochemical assays using frog egg extracts at constant temperatures (here for $T = 24\,°C$), see Supplementary Note 6. **B** The assays were repeated for temperatures in the interval $16–26\,°C$, and (apparent) activation energies were extracted. Blue dots represent data of individual fitted time series, while the orange dots are the medians per temperature. **C** Probability distribution of fitted (apparent) activation energies using bootstrapping with 90% confidence intervals (see Supplementary Note 2 for details on the bootstrap procedure). Source data are provided as a Source Data file[96,98].

measurements of the two processes in *X. laevis* frog egg extracts[86] (Fig. 6A, B). The synthesis of one mitotic cyclin, cyclin B2, in cycling extracts was monitored by quantitative Western blotting, using the cyclin B2 levels present in CSF extracts as a normalization standard (Fig. S14). APC/C activity was gauged by introducing securin-CFP, translated in wheat germ extracts, as a fluorescent reporter of APC/C activity into CSF extracts, then driving the extracts out of CSF arrest with calcium plus cycloheximide and into mitotic arrest with non-degradable cyclin B (Fig. S15). Experimental protocols are detailed in Supplementary Note 6. These measurements were conducted across temperatures ranging from 16 °C to 26 °C and did not extend into the high temperature range where the rate of cyclin synthesis as inferred in Fig. 5B began to drop. These rate data were consistent with the Arrhenius equation (Fig. 6B, green line), and cyclin synthesis was more sensitive to temperature than cyclin degradation, with fitted apparent Arrhenius energies of 87 and 51 kJ/mol, respectively. Bootstrapping supported the statistical significance of this difference (Fig. 6C). This provides direct support for the hypothesis that the different scaling of opposing enzymes contributes to the non-Arrhenius character of the cell cycle period.

We also measured the temperature dependence of two other key cell cycle regulators, cyclin B-Cdk1 and PP2A-B55 (Figs. S16, S17). These opposing enzymes are critical for the phosphorylation and dephosphorylation of many cell cycle proteins, and their activities would be expected to contribute to the dynamics of mitotic entry and mitotic exit. Figure 6 suggests minor variations in their temperature sensitivity, with apparent Arrhenius energies of 46 and 56 kJ/mol, compatible with robust functioning of the cell cycle oscillator over its nominal temperature range.

## Decreasing the cyclin synthesis rate decreases the viable temperature range

Our analysis suggests that the failure of the *Xenopus* embryonic cell cycle oscillator at temperature extremes is governed by two distinct mechanisms. At high temperatures, oscillations break down due to the biphasic temperature dependence of cyclin B synthesis: the synthesis rate declines with increasing temperature and eventually becomes insufficient to sustain oscillations in the face of baseline degradation. At low temperatures, failure arises from an imbalance in Arrhenius scaling. Specifically, the higher activation energy of synthesis relative to degradation causes the synthesis rate to become too low to counteract degradation. Thus, the temperature range over which the oscillator functions is determined by the temperature-dependent

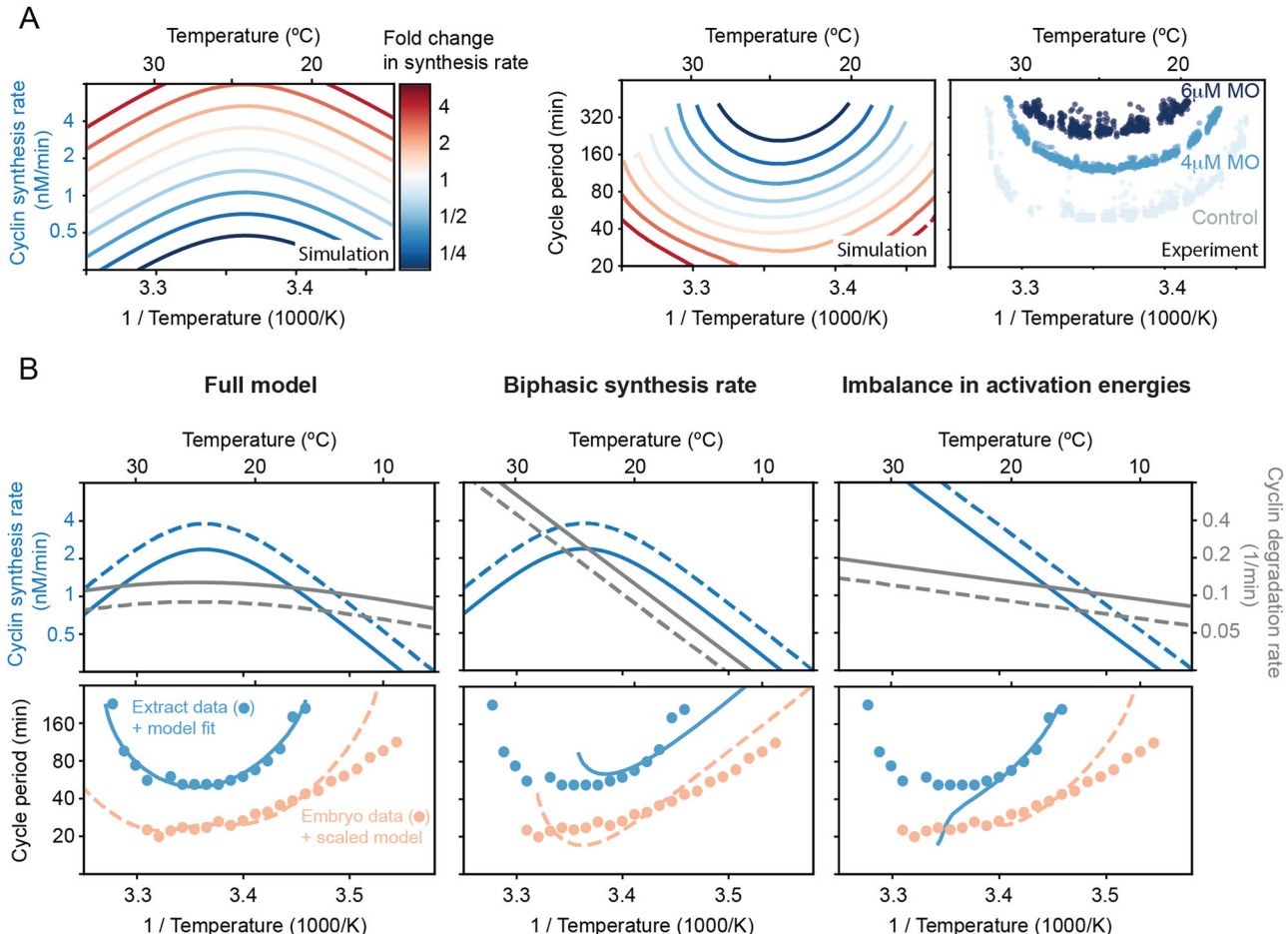

**Fig. 7 | Decreasing the cyclin synthesis rate decreases the viable temperature range. A** Influence of changing the basal cyclin synthesis rate by a factor up to 5 on the shape of the temperature response curves. The degradation rate is scaled up to a factor of 3. The two left panels show simulations of the 2-ODE model of the cell cycle oscillator using a parameter set obtained from the ABC method (one of the gray lines in Fig. 5B), plotting the temperature-dependence of the cyclin synthesis rate and the corresponding cell cycle period. The right panel shows the cell cycle duration as a function of temperature obtained from encapsulated extracts with 0, 4, or 6 μM morpholino (MO) oligonucleotides against isoforms of Xenopus cyclin B1/B2 mRNA species, thus lowering the cyclin synthesis rate. **B** Different scenarios in temperature dependence of cyclin synthesis and degradation lead to different non-Arrhenius scaling of cell cycle oscillations. While a biphasic cyclin synthesis rate leads to a double exponential response curve, the imbalance in activation energies introduces a curved non-Arrhenius response at lower temperatures, which is critical for reproducing the experimental data measured in frog egg extract. Source data are provided as a Source Data file[98].

interplay of these two opposing processes, both essential for cell cycle progression.

This reasoning predicts that modulating the overall cyclin synthesis rate should systematically alter the temperature range over which the oscillator can operate. To test this, we used our 2-ODE model of the cell cycle oscillator using a parameter set obtained from the ABC method (one of the gray lines in Fig. 5B). We then systematically varied the synthesis rate at a reference temperature while preserving its temperature dependence. At the same time we allowed the degradation rate to be scaled similarly, but to a lesser extent. In Fig. 7A, we show one representative example (see more simulation details in Fig. S18). The resulting temperature-period curves were U-shaped and shifted in a consistent, and perhaps non-intuitive, way: increasing the synthesis rate led to faster oscillations and broader temperature ranges, while decreasing it caused both upper and lower temperature bounds to move inward. At high synthesis rates, oscillations failed, starting at intermediate temperatures.

We next tested this prediction experimentally by titrating cyclin B morpholino antisense oligonucleotides (0, 4, or 6 μM) into encapsulated extracts subjected to a temperature gradient (see Materials and "Methods"). These morpholinos inhibit cyclin B translation by binding to its mRNA. As predicted, increasing morpholino concentrations

resulted in longer minimum cycle periods and a narrower viable temperature range (Fig. 7A), consistent with the model. A quantitative discrepancy remained, with experimental oscillations persisting at more extreme temperatures than predicted. This could reflect additional regulatory layers that are not captured in our minimal model, beyond cyclin synthesis/degradation and Cdk1 (in)activation. Moreover, the cell cycle oscillator in extracts is also gradually slowing down over time (see Fig. 4B), which could also explain the discrepancy between the experiments and the idealized, time-invariant simulations.

We then asked whether the same model, tuned to cycling extract data, could recapitulate the cell division timing in early *X. laevis* embryos (Fig. 7B). While the general shape was similar, oscillations in embryos were systematically faster than in extracts. This could be due to cytoplasmic dilution during extract preparation—though such effects are modest at moderate dilutions[87]—or the absence of nuclear and membrane components. The period differences between embryos and extracts were also similar to variations among extracts from different biological replicates (Fig. S7). We therefore rescaled the cyclin synthesis by a factor of 1.6 and the degradation rates by 0.7 to match the cycle duration at 25 °C. Remarkably, this single-point scaling allowed the model to capture the embryo temperature response curve with reasonable accuracy across a broader temperature range than

would be achieved by simply scaling the extract response curve. (Fig. 7B, bottom left). The embryo data did deviate at lower temperatures, suggesting additional embryo-specific dynamics not captured by our simple model.

Finally, we asked whether either mechanism, biphasic synthesis or an imbalance in Arrhenius scaling exponents, could alone account for the observed temperature dependence. Using our calibrated model, we isolated each effect and found that both independently produce non-Arrhenius temperature scaling (Fig. 7B, middle-right). However, neither mechanism alone reproduced the fine structure of the experimental curves, indicating that both are required to explain the wide temperature adaptability of the embryonic cell cycle oscillator. In particular, a biphasic cyclin synthesis rate leads to a double exponential response curve, capturing deviations at high temperatures, but still appears Arrhenius-like at low temperatures. Especially for the extract data, such a model fails to capture the strong low-temperature deviations. In contrast, the imbalance in activation energies introduces a curved non-Arrhenius response across all temperatures, yet it fails to capture the sharp increase in period at high temperatures.

Taken together, these findings support our model, which attributes the non-Arrhenius scaling properties of the early embryonic cell cycle oscillator to two concurrent mechanisms. At low temperatures, the scaling is primarily driven by an imbalance between opposing cyclin synthesis and degradation rates. In contrast, at high temperatures, the biphasic nature of cyclin synthesis plays a critical role in capturing the upward curvature of the oscillation period.

## Discussion

Previous work suggested that the early embryonic cell cycle scales similarly with temperature in several organisms[30–33]. Here we have extended these measurements to *Xenopus tropicalis* and *Danio rerio*, and have supplemented previous work on *Xenopus laevis* with additional types of measurements. We found that although the periods of the cell cycle at the organisms' nominal temperatures vary from about 5 min for *C. elegans* and *C. briggsae* to about 25 min for *X. laevis*, the temperature scaling of the periods is quite similar. The apparent Arrhenius energies averaged $75 \pm 7$ kJ/mol (mean ± std. dev., $n = 6$), and the average $Q_{10}$ value at 20 °C was $2.8 \pm 0.2$ (mean ± std. dev., $n = 6$) (Figs. 1, 2). In all cases the periods deviated from the Arrhenius relationship at high temperatures, and for *X. laevis*, the Arrhenius plots were non-linear throughout the range of permissible temperatures.

In some ways it is perhaps not surprising that the temperature scaling data could be approximated reasonably well by the Arrhenius equation. Crapse et al.[32] have shown computationally that chaining together a sequence of chemical reactions results in only minor deviations from ideal Arrhenius scaling if one assumes that the individual enzymes' activation energies do not differ greatly. Experiments have shown that Min protein oscillations, crucial for bacterial cell division, also display Arrhenius-like scaling behaviors[88]. The classic chemical oscillator, the Belousov-Zhabotinsky reaction, approximately obeys the Arrhenius equation[89–91], and in general, many biological processes at least approximately conform to the Arrhenius equation or one of the proposed modified versions of the Arrhenius equation[15–22].

These observations notwithstanding, it was not obvious to us why a complex oscillator circuit, with non-linearities and feedback loops, should yield even approximately Arrhenius temperature scaling, and what the origins of the experimentally observed deviations from Arrhenius scaling might be. Through modeling studies, we identified two plausible mechanisms for the observed non-Arrhenius behavior: an emergent mechanism resulting from differences in the Arrhenius energies of opposing enzymes in the network (Fig. 3D, Case 3), and a biphasic temperature dependence for one or more of the critical individual enzymes (Fig. 3D, Case 4). A priori, either or both of these mechanisms could pertain.

Experimental observations combined with model-based inference suggest that a key step in the oscillator circuit − the synthesis of the mitotic cyclin protein − exhibits a strongly biphasic dependence on temperature. While intact *Xenopus* embryos do not survive above 29 °C, cell-free cycling extracts can continue oscillating at temperatures exceeding 30 °C. Above approximately 30 °C, the rate of cyclin synthesis (as inferred from the Cdk1 activity sensor) and the rate of progression through interphase clearly decrease with increasing temperature, whereas at lower temperatures they increase with increasing temperature (Fig. 4). Our hypothesis is that above some maximum permissible temperature, the imbalance between the cyclin synthesis and degradation rates causes the oscillator to fail and the cell cycle to arrest.

Cyclin synthesis and degradation also scaled differently with temperature at the low end of the permissible temperature range. This was inferred from fitting the parameters of the two-ODE model to the experimental data (Fig. 5), and then was directly shown by in vitro assays of cyclin synthesis and degradation (Fig. 6). This means that below a critical temperature, cyclin synthesis and degradation should again be out of balance, causing oscillations to cease.

To further test this hypothesis, extracts were treated with a mixture of four morpholino oligonucleotides, two for cyclin B1 and two for cyclin B2, to inhibit cyclin translation enough to slow but not block the cell cycle at normal temperatures (see Materials and Methods). We asked whether this decreased the maximum permissible temperature, raised the minimum permissible temperature, or both. We found that both temperature limits were similarly affected, and the operating range of the cell cycle oscillator was narrowed, as predicted by our simple two ODE model (Fig. 7). This finding is consistent with the hypothesis that both operating limits are determined by the balance between cyclin synthesis and degradation.

One question then is, why did evolution not arrive at a system where cyclin synthesis and degradation did not go out of balance, at high and low temperatures? We suspect that there are trade-offs between competing performance goals for the oscillator and its components. Perhaps the molecular flexibility required to make protein synthesis run as fast as possible at the temperatures typically experienced by an ectotherm render the ribosomes vulnerable to unfolding at slightly higher temperatures. Likewise, cyclin synthesis and degradation might work best at normal temperatures even if their temperature scaling does not perfectly match the overall system's activation energy, suggesting that the observed $E_a$ reflects a trade-off between fast reaction rates and ideal scaling.

While our study focuses on early embryonic systems that are largely transcriptionally silent, recent work in yeast[92] has shown that temperature-induced changes in gene expression can drive fate decisions, and synthetic bacterial circuits have been engineered to achieve temperature compensation through specific protein modifications[93]. These findings highlight complementary mechanisms of thermal adaptation, from network-level emergent scaling, as we demonstrate here, to dedicated molecular adaptations. Furthermore, synthetic gene circuit evolution studies[94] offer promising opportunities to explore how temperature robustness can arise in engineered systems, providing a future experimental platform to test and extend the principles uncovered here.

One final question is how the behaviors seen here compare to those of the same circuit in endotherms, organisms that have at great metabolic cost freed their biochemistry from needing to function reliably over such wide temperature ranges. Although the four enzymatic processes individually assessed here (cyclin synthesis, cyclin degradation, Cdk1 activity, and PP2A-B55 activity) differed in their temperature scaling, they did not differ by that much; their $E_a$ values averaged to 60 kJ/mol with a standard deviation of 16 kJ/mol or 27%. It seems plausible that endothermy might allow enzymes with a wider

range of activation energies to be used than would be possible in ectotherms.

## Methods

### Animal care

For the embryo experiments, the Ethical Committee for Animal Experimentation KU Leuven approved the frog handling, including the injections and the egg collection, under Project 107-2021. All frog colonies were housed in recirculating systems with temperature and water quality regularly monitored. For the extract experiments, all experiments and animal procedures were conducted in compliance with ethical regulations and the description in the Institutional Animal Care and Use Committee (IACUC) approved protocol (PRO00011571) at the University of Michigan-Ann Arbor.

### Xenopus egg extract

Cell-free cycling extracts and CSF extracts were made from *Xenopus laevis* eggs following a published protocol from Murray[83]. For cycling extracts, this protocol was adapted as in[81]. Extracts for Fig. 3B–G and Fig. 4 were then supplemented with 1 µM Cdk1-FRET sensor, as described in Maryu and Yang[78], and also with 1X energy mix (7.5 mM Creatine phosphate, 5 mM ATP, 1 mM EGTA, 10 mM MgCl2). Work from the Yang lab demonstrated that an intermediate range of dilution of the extracts can improve the number of cycles, with the best activity at around 30% dilution[87]. As a result, for the data described here, the dilution was kept constant at 30% with extract buffer (100 mM KCl, 0.1 mM CaCl2, 1 mM MgCl2, 10 mM potassium HEPES, 50 nM sucrose, pH 7.8). Extracts for the biochemical assays in Fig. 4A were undiluted.

The extract was encapsulated via a water-in-oil emulsion using a microfluidic device. The fabrication of the device and droplet generation followed a previously published protocol[95]. Briefly, cycling extract (water phase) was mixed with 2% 008-FluoroSurfactant in HFE7500 (Ran Biotechnologies, Inc.) (oil phase) inside a microfluidic device driven by an Elveflow OB1 multi-channel flow controller (Elveflow). Air pressure was 2 psi for both the extract and oil channels. After droplets were generated, they were loaded into VitroCom hollow glass tubes with a height of 100 µm (VitroCom, 5012) pre-coated with tri-chloro (1H,1H,2H,2H-perfluorooctyl) silane, and then immersed in a glass-bottom dish (WillCo Wells) filled with heavy mineral oil (Macron Fine Chemicals) to prevent evaporation.

### Temperature gradient generation

A custom plastic microscope stage was fabricated to fit two aluminum plates on each side of the imaging dish. Each aluminum plate was attached to a TEC1-12706 40*40MM 12V 60W Heatsink Thermoelectric Cooler Cooling Peltier Plate (HiLetgo) using thermal conductive glue (G109, GENNEL). The plate designated for temperatures above room temperature had an additional heatsink (40 mm x 40 mm x 20 mm, black aluminum, B07ZNX839V, Easycargo) and a cooling fan to improve performance. The plate designated for cold temperatures had an additional liquid cooling system (Hydro Series 120 mm, CORSAIR) attached with thermal conductive glue.

Peltier devices were controlled via two CN79000 1/32 DIN dual-zone temperature controllers (Omega). In all experiments, the target temperature was set to 65 °C and 1 °C for the hot and cold plates, respectively. With both plates on, it was always the case that the hot plate reached its target temperature and stayed constant within 5–10 min, and the cold plate stayed stable at 10 °C.

The imaging dish was attached to the aluminum plates with Thermal adhesive tape 2-5-8810 (DigiKey) to ensure proper thermal conduction. Temperature was logged via 4 K-Bead-Type thermo-couples placed on the imaging dish touching the bottom surface. Data was acquired using a 4-channel SD Card Logger 88598 AZ EB (AZ Instruments). Room temperature was also captured using the same method via a thermocouple attached to the microscope stage.

### Western blotting

Cycling extracts were prepared according to the method by Murray et al.[83], except that eggs were activated with calcium ionophore A23187 (5 µl of a 10 mg/ml stock of A23187 in 100 ml 0.2x MMR) rathgrher than with electric shock. After preparing the extracts, they were distributed to several eppendorf tubes and brought to a specified temperature between 16 and 26 °C within 20 min. 2 µl samples were taken every 4 min (every 8 min at 16 °C) and immediately frozen on dry ice. To each 2 µl aliquot, 48 µl of SDS sample buffer supplemented with DTT was added, and the samples were boiled during 10 min at 95 °C. 12 µl of the cycling extract samples and 4 µl of the reference samples (CSF extract prepared according to the method by Murray et al.[83], were run on 10% Criterion TGX Precast protein gels and transferred to a PVDF membrane using the Bio-Rad Trans-blot Turbo system. After blocking in milk (4% w/v in TBST), the blots were incubated with a 1/500 dilution of anti-cyclin B2 antibody (X121.10, Santa Cruz) overnight at 4 °C followed by a 1/10.000 dilution of anti-mouse IgG HRP-linked whole secondary antibody (GE Healthcare NA931), during 1 hour at room temperature. Finally, the blots were developed using Supersignal West Femto che-miluminescent substrate. The images of the blots can be found on the Zenodo repository[96].

### H1 kinase activity assay

H1 kinase assays were performed following the protocol described in[86]. Briefly, 2 µL of frozen extract was diluted in 98 µL of EB buffer (80 mM $\beta$-glycerophosphate, 20 mM EGTA, 15 mM MgCl, pH 7.4). Ten micro-liters of this diluted extract were then combined with 10 µL of reaction buffer containing: 20 mM HEPES (pH 7.5), 5 mM EGTA, 10 mM MgCl, 200 mM ATP, 10 µg histone H1 (Millipore, #14-155), 20 µM PKA inhi-bitor IV (Santa Cruz Biotechnology, #sc-3010), and 2.5 µCi [$\gamma$-$^{32}$P]-ATP. Reactions were immediately incubated for 3 min at 20 °C. Reactions were stopped by adding 20 µL of 3 × SDS-PAGE loading dye. Five microliters of each sample were loaded onto a 10% Criterion Tris-HCl precast gel (Bio-Rad, #3450011), separated by electrophoresis, trans-ferred to a PVDF membrane, and dried. Radiolabeled histone H1 was visualized using a BAS Storage Phosphor screen (GE Healthcare) and imaged on a Typhoon 8600 Phosphorimager (Molecular Devices).

### APC/C activity assay

APC/C activity assays were performed as described previously[66]. Briefly, securin-CFP was synthesized by in vitro translation using the TNT SP6 High-Yield Wheat Germ Protein Expression System (Promega, Cat. No. L3261). For each reaction, 3–4 µL of translated securin-CFP was added to 50–80 µL of Xenopus egg extract. The mixture was divided into 20 µL aliquots and transferred to a black 384-well plate (Greiner, Cat. No. 781076). Fluorescence of securin-CFP was monitored in real time using a FlexStation II plate reader (Molecular Devices). Degradation rates were calculated by normalizing fluorescence values to the starting intensity and background controls, followed by fitting a single exponential decay curve. All measurements were performed in duplicate or triplicate.

### PP2A-B55 activity assay

PP2A-B55 activity assays were performed as previously described[97] with slight modifications. In brief, maltose-binding protein fused to amino acids 38 to 62 of *Xenopus laevis* Cdc20L (Fizzy) was recombi-nantly expressed in E. coli and affinity-purified using amylose resin. Without elution, about 1 mg of protein was phosphorylated in 200 µL kinase reaction buffer (20 mM HEPES pH 7.7, 10 mM MgCl2, 15 mM KCl, 1 mM EGTA, 5 mM NaF, 20 mM $\beta$-glycerophosphate, 10 µM ATP, 2.5 µg cyclin A2/CDK2 (Sigma, C0495), 60 µCi [$\gamma$-$^{32}$P]-ATP) overnight at 37 °C. The resin was extensively washed, and the labeled substrate eluted with elution buffer (20 mM Tris-HCl pH 7.5, 150 mM NaCl, 10 mM maltose). The substrate was concentrated and stored at −20 °C until use. For measuring PP2A-B55 activity, 1 µL of substrate (>15,000 cpm)

was added to 5 μL of extract and incubated for 12 min at 20 °C (3 min for the time course measurements). The reaction was stopped by adding 20 μL 10% ice-cold TCA and stored on ice until further processing. Samples were spun for 10 min at 14,000 $g$, and 20 μL of the supernatant was transferred to a fresh tube. Thirty microliters of 5% ammonium molybdate in 0.5 M sulfuric acid was added and mixed. Fifty microliters of water-saturated heptane/butanol was added and the solution vortexed for 30 s. The solution was spun for 10 min at 14,000 $g$, and 30 μL of the organic upper phase was used for the detection of the inorganic phosphate using a scintillation counter.

## Time-lapse fluorescence microscopy

For Figs. 4, 5, 7, imaging was carried out on an inverted Olympus IX83 fluorescence microscope with a 4 × air objective, a light-emitting diode fluorescence light source, a motorized x-y stage, and a digital complementary metal-oxide-semiconductor camera (C13440-20CU, Hamamatsu). The open-source software μManager v1.4.23 was used to control the automated imaging acquisition. Bright-field and multiple fluorescence images of CFP, FRET, and YFP were recorded at a frequency of one cycle every 3 to 7 min for 40 to 50 h for each sample.

## Image processing and analysis methods

Grids of images were captured and subsequently stitched together using ImageJ's Grid/Pairwise Stitching plug-in, in conjunction with additional pipeline code written in Fiji/Java. Bright-field images from the first frame were used to generate stitching parameters, which were fed to ImageJ to stitch each channel at each frame consecutively. The FRET ratio was calculated as in Maryu and Yang[78].

For Figs. 4, 5, 7, custom scripts in MATLAB 2020a and ImageJ were written to perform image processing. Briefly, each microscope position was processed by manually selecting the region containing the tube of interest and then algorithmically cropping and resizing that region in all channels. Then, bright-field images were used for individual droplet segmentation and tracking using Trackmate 7.12.1. Only individual droplets whose radius was smaller than 100 μm and track started within the first 60 min of the experiment were selected for further analysis. FRET ratio intensity peaks and troughs were first auto-selected and then manually checked and corrected using custom Python scripts. Rising and falling periods were calculated from this data. All code is available at ref. 98. The tracking data can be found on the Zenodo repository[96].

## Morpholino oligonucleotides

A combination of four morpholino antisense oligonucleotides (Gene Tools, LLC) at equal concentrations was designed, and their sequences are as follows:
- Morpholino anti-*Xenopus*-CyclinB1 (ccnb1_a): ACATTTTCCCAAAACCGACAACTGG
- Morpholino anti-*Xenopus*-CyclinB1 (ccnb1_b): ACATTTTCTCAAGCGCAAACCTGCA
- Morpholino anti-*Xenopus*-CyclinB2 (ccnb2_l): AATTGCAGCCCGACGAGTAGCCAT
- Morpholino anti-*Xenopus*-CyclinB2 (ccnb2_s): CGACGAGTAGCCATCTCCGGTAAAA

We applied a total concentration of 0, 4, or 6 μM of the morpholino cocktail to the cycling *Xenopus* extracts to suppress the endogenous translation of cyclin B1/B2. These concentrations were chosen within a dynamic range that could inhibit cyclin translation but should not terminate the cell cycle at normal temperatures, based on microfluidic channel tuning experiments[99].

## Fitting of scaling laws

For the fitting of Arrhenius and other functional forms, we always binned the data per integer temperature value and then took the median per temperature. This results in a dataset with one rate/duration per temperature, which is the basis for the fits.

For fitting the Arrhenius equation, we use linear regression on the logarithm of the duration $\Delta t$ and $1/T$ where $T$ is the absolute temperature.

We also fitted a *Double Exponential* (DE) function, which contains two exponential functions and four free parameters:

$$\Delta t = A_1 e^{\frac{-E_{a1}}{RT}} + A_2 e^{\frac{-E_{a2}}{RT}}. \tag{5}$$

To fit the four parameters, we use a two-step approach. First, on a manually selected Arrhenius interval, we fit the standard Arrhenius law. This yields fitted values of the duration $\hat{P}$. Next, we take the durations for the other temperatures and fit an Arrhenius law on the differences $P - \hat{P}$, such that the sum of these fits describes the whole curve. Next, we used the resulting parameters as starting values in a full nonlinear fit of Eq. (5) using the `curve_fit` function from `scipy`.

The *Quadratic Exponential* (QE) function, which contains three free parameters, is given by

$$\Delta t(T) = A e^{\frac{-E_a}{R}\left(\frac{1}{T} + \frac{B}{T^2}\right)}. \tag{6}$$

It can be fit using standard least squares on the logarithm of the duration as a function of $1/T$.

The *Power law - Exponential* (PE) function is given by

$$\Delta t(T) = A T^B e^{\frac{-E_a}{RT}}. \tag{7}$$

The fit is done using standard least squares. The code for fitting the functions is all included in the GitHub repository.

Note that for the fitting of the data for *X. tropicalis* embryos, we left out the point at the lowest temperature since this seems to be an outlier.

The bootstrap procedure we used to obtain distributions for the fitted activation energies is described in detail in Supplementary Note 2.

## ODE modeling

The equations and parameter values for both the 2 ODE model and the 5 ODE model are described in Supplementary Note 3. Simulations were performed in Python using `solve_ivp` from the `scipy` package. In general, we simulated for a time of 1000 min using the `BDF` solver.

To detect the cycle period from a simulation, we detect peaks in the timeseries of the cyclin variable, and use the last two peaks to determine the period. If these peaks are too different in their y-values, we don't consider the system oscillating, as this would correspond to a damped rather than a sustained oscillation.

## Reporting summary

Further information on research design is available in the Nature Portfolio Reporting Summary linked to this article.

## Data availability

Source data are provided with this paper. • The western blots and droplet tracking data have been deposited in a Zenodo repository[96], publicly available as of the date of publication. • All datasets necessary to reproduce the figures in the manuscript have been deposited in a Zenodo repository[98], publicly available as of the date of publication. Source data are provided with this paper.

## Code availability

Codes are provided with this paper. • All datasets and original modeling codes necessary to reproduce the figures in the manuscript have been deposited in a Zenodo repository[98], publicly available as of the

date of publication. • All codes for image processing and analysis methods have been deposited in a Zenodo repository[98], publicly available as of the date of publication.

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

## Acknowledgments

The work is supported by grants from the National Institutes of Health (R01 GM046383 and P50 GM107615, J.E.F., R01GM144584, Q.Y.), the National Science Foundation (MCB#2218083, Q.Y.), Internal funds KU Leuven (C14/23/130, L.G.), a junior research grant from the Research Foundation - Flanders (G074321N, L.G.), a doctoral fellowship from the Research Foundation - Flanders (11D0918N, J.R.) and postdoctoral fellowships from EMBL/EIPOD4 (Marie Skłodowska-Curie Cofund actions 847543, J.R.) and FNRS (Chargé de recherche, 40024839, J.R.). We acknowledge the support of the EMBL HPC resources. We thank Ernesto Flores for his contributions to the design and testing of the temperature chamber during his NSF REU project in the Yang lab in the summer of 2022.

## Author contributions

L.G., Q.Y., and J.E.F. conceived the study; F.T., A.V., C.P., and L.G. conducted the experiments; J.R., F.T., and L.G. analyzed the data; J.R. developed and analyzed the models; J.R., F.T., J.E.F., and L.G. prepared the figures; L.G. and J.E.F. wrote the manuscript, with L.G. incorporating feedback from all authors.

## Competing interests

The authors declare no competing interests.
