## [Peer Review file · Nature Communications]

Mechanistic origins of temperature scaling in the early embryonic cell cycle

Corresponding Author: Professor Lendert Gelens

Version 0:

Reviewer comments:

Reviewer #1

(Remarks to the Author)

The manuscript by Rombouts et al addresses the impact of temperature on the embryonic cell cycles. This is an important biological question especially in embryos that develop outside as they must experience different temperatures during development. This topic has been addressed by others, but this paper presents a more detailed analysis of the cell cycle oscillator thus making interesting contributions. Specifically, the analysis of the dependence on temperature of different cell cycle regulatory aspects is interesting. Overall, I think that this paper is a good contribution to the field. Here are some comments that the authors should consider to improve the paper.

Fundamental comments:

1. Why is “cyclin synthesis” estimated in Figure 5B biphasic but not when measured in Figure 6B? I can imagine ways to reconcile this, but the big problem is that what is estimated as “cyclin synthesis” in Figure 5B could potentially correspond to or be influenced by other molecular processes. What the authors are showing is the rate of increase of the phosphorylation of a Cdk reporter. This could be influenced by the rate of inactivation of mitotic phosphatases, by some other factors that control Cdk activity independently of cyclin synthesis, etc. With available data, I am not convinced that the author can be confident of the biphasic nature of cyclin synthesis. I am also confused on how to reconcile the data on the duration of falling phase in Figure 4E, the data on cyclin degradation in Figure 5B and the dependency of APC and PP2A activity on temperature. The dependencies on temperature are very different, how do I square all this?
2. The prediction for Figure 7A suggests a much bigger effect of reduction of cyclin synthesis than it is experimentally observed. In the model, oscillations would stop before the cycle reaches a duration of ~300 min, while in experiments oscillations of about 300 minutes are observed from 20-30 C when 6uM of Morpholinos are used. This links to the worry that cyclin synthesis is not the main or only determinant of the increase in Cdk activity. I think that the fact that there is an effect is interesting, but the authors are over-interpreting their results. Similarly in Figure 7C the gymnastic used to get a good fit of the model is not fully justified and the fit is actually not great with systematic deviations which are not properly acknowledged. Overall, I feel that the evidence for the biphasic nature of cyclin synthesis is weaker than the authors claim. That does not significantly impact my evaluation of the paper, but the authors need to present a more balanced conclusion. I would suggest that the author say that there is a biphasic dependency on temperature of the rate of Cdk activation in interphase and that the rate is partially but not fully controlled by the rate of cyclin synthesis.

Major comments:

3. While I think that it is the authors' prerogative to write the paper they want, I did not enjoy reading this paper, as there was a lot of text explaining things that are either very basic or irrelevant to the flow. Just few examples: pages 3-5 have lengthy explanations of Arrhenius' law, a lengthy and pedantic explanation of Q10 which is disconnected from the rest of the paper, a lengthy explanation of how to fit double exponentials which could be moved to the methods (also I highly doubt that this is a paper where the amount of data required efficient fitting strategies and fitting two exponentials to data is not a computationally intensive procedure).
4. The extensions of the Arrhenius law and the fact that they fit data better is somewhat trivial as all the extensions are models with extra parameters and there is no attempt to correct for that. Furthermore, the insights from those fits are not relevant to understand the rest of the paper, so I found that the whole session adds little value to the paper and could be removed. To be clear, I feel that the authors should have the freedom to write the paper that they want to write and not the one that I would write, so it is up to them whether to follow this suggestion and the previous one. However, I want to make the authors aware that, as a reader, by the time I got to the insightful stuff I had some fatigue from having to read through

things that did not seem very insightful or relevant.

5. The comments on *Drosophila* on page 4 are wrong. The cell cycle data the authors analyze is cycle 13 and there is no G2 in cycle 13. It is a longer cell cycle due to lengthening of S-phase but no G2 phase yet. Cycle 14 is asynchronous and lasts at least 75 min at room temperature. Similar data with temperature scaling at cycle 13 and with Cdk1 measurements are also present in Hayden et al *Current Biology* 2022 and could be cited and added to the authors' analysis.

6. The authors present two models for Cdk1 oscillations: a model dominated by positive feedback via Cdk1/Wee1/Cdc25 and a model dominated by the Cdk1/Greatwall/PP2A feedback. They then show that similar conclusions can be obtained on the dependency of temperature, that is the observed data require a biphasic dependency of synthesis and different temperature dependency. Interestingly, they also comment that the second model applies to cycle 2-12 in vivo, while the first model applies to the first cycle and in vitro cycles (I am extrapolating here, so please be clearer about which regimes the two models describe for the readers' information). It is interesting that the main "theoretical" conclusion of the paper is independent of the model used, but I also wonder whether modeling is even required and whether one could have reached this conclusion in a model-independent manner, as it seems that modeling adds very little in a way. This is not a major concern to me, as I think that the main strength of this paper is in the experimental analysis which can be rationalized in a model-independent manner. This is just to say that I found the modeling part not very insightful. Again up to the authors to decide what to make of my comment here.

7. I don't find Figure 4E very helpful. What is the point of defining an activation energy for a system that clearly does not behave like Arrhenius? What does activation energy even mean then? Since the authors use activation energies later in Figure 6 for related molecular events, it is also a bit unclear how the data in Figure 4E should be interpreted. Maybe I missed something, but I was confused- see also my first point.

8. The analysis of Cdk activity and the additional biochemical experiments to get the different rates and their dependency on temperature is very interesting and the major contribution of this paper. The only minor comment I have is that the use of the Cdk1 sensor was inspired by work from Pines who made the first Cdk sensor and work from Di Talia who demonstrated the first use of the sensor in embryos, yet none of those papers are cited and only the paper from the Yang's lab that made the sensor specifically used here is cited. I think that it would be best to cite the papers that inspired making this tool in *Xenopus*.

(Remarks on code availability)

Reviewer #2

(Remarks to the Author)

This manuscript investigates the temperature-dependence of the *Xenopus laevis* cell cycle at various levels by experiments in fertilized oocytes, oocyte extracts, and by mathematical/computational modeling with ODEs and nonlinear dynamics. Valuable comparisons with fish and worm embryos provide insight into the generality of the observations. The findings indicate Arrhenius scaling over a range of temperatures for embryo cell division timings and other post-fertilization events. While the actual timings may be different, the activation energies are remarkably similar across events and species. A simple and elegant mathematical model leads to a relaxation oscillator that matches the essence of embryo cell cycle dynamics with three-phase periods. Examining the 4 rates in the model leads to the conclusion that either independent temperature-scaling or biphasic scaling of at least one parameter's temperature-dependence explains the deviations in cell cycle timings from the Arrhenius model outside of the good-fit temperature range. Further experiments highlight that both independent and biphasic scaling are at play, with cyclin synthesis being more temperature-sensitive than cyclin degradation, and differences in the temperature-scaling of interphase and M-phase durations.

This is a great interdisciplinary manuscript that was a joy to read both from a theory and experimental perspective. It provides many significant insights into fundamental questions that also have environmental relevance in a changing climate. The work is well executed and supports the conclusions. I would like to recommend publication after the authors address the following comments.

- (1) The investigation of similarities between different animal species gives nice insights. It would be interesting to expand more. Since yeast is mentioned, how would these results generalize to eukaryotes? Previous work showed a cell fate decision at high temperatures in yeast, which would be worth discussing, see PMID:30341217. Another interesting reference might be on the temperature-compensation of a synthetic bacterial oscillator, see PMID:24395809.
- (2) As exposed in PMID:30341217, temperature affects cells at multiple scales, including gene expression and protein structure. It would be interesting to discuss how gene expression alterations due to the temperature-dependence of mRNA and protein synthesis and degradation/dilution for the cell cycle machinery, including Cdk1, Cdc25, Wee1 could affect the model – which is mostly at the protein interaction/activity level.
- (3) In Fig. 1, how does the period between fertilization and fertilization wave completion depend on temperature? It would be interesting to know, since the other periods are all plotted.
- (4) Unlike the rest of the paper, the last section before the Discussion is somewhat unclear. Can this be more clearly written? What are morpholino oligonucleotides and how do they help address the question in this section?
- (5) How would these temperature-dependences act in adult animals with full cell cycle and developed tissues? Any implications for human biology? Generality for genetic oscillators? There are interesting and possibly relevant observations for p53 oscillations, see PMID:32001771.
- (6) Are there any clues whether the imbalanced temperature-sensitivity of cyclin synthesis versus degradation could be conserved in other animals, or even yeast?
- (7) What are the implications of these findings on designing or evolving synthetic genetic oscillators? A relevant reference would be PMID:38925113 when discussing this aspect.

(8) Figure 2B: Maybe indicate the optima by an arrowhead or some other symbol.

(Remarks on code availability)

Unfortunately access to the code is restricted.

It seems like only KU Leuven faculty or staff members or students can access the code.

Version 1:

Reviewer comments:

Reviewer #1

(Remarks to the Author)

The authors have addressed my main comments by rebalancing their interpretations of few crucial experiments and conclusions. I think that this nice paper can be accepted in its current form.

(Remarks on code availability)

Reviewer #2

(Remarks to the Author)

(Remarks on code availability)

The code is transparent and usable for the community.

Response to Reviewers

Reviewer #1 (Remarks to the Author):

The manuscript by Rombouts et al addresses the impact of temperature on the embryonic cell cycles. This is an important biological question especially in embryos that develop outside as they must experience different temperatures during development. This topic has been addressed by others, but this paper presents a more detailed analysis of the cell cycle oscillator thus making interesting contributions. Specifically, the analysis of the dependence on temperature of different cell cycle regulatory aspects is interesting. Overall, I think that this paper is a good contribution to the field. Here are some comments that the authors should consider to improve the paper.

We thank the reviewer for the feedback. We are glad to hear our detailed analysis of the cell cycle oscillator under varying temperatures is a valuable contribution. We appreciate the comments and carefully address them below and in the manuscript.

Fundamental comments:

1. Why is “cyclin synthesis” estimated in Figure 5B biphasic but not when measured in Figure 6B? I can image ways to reconcile this, but the big problem is that what is estimated as “cyclin synthesis” in Figure 5B could potentially correspond to or be influenced by other molecular processes. What the authors are showing is the rate of increase of the phosphorylation of a Cdk reporter. This could be influenced by the rate of inactivation of mitotic phosphatases, by some other factors that control Cdk activity independently of cyclin synthesis, etc. With available data, I am not convinced that the author can be confident of the biphasic nature of cyclin synthesis.

Response: The data in Figure 6B were measured over a more limited range of temperatures. The experiments did not get into the high temperature range where the rate of cyclin synthesis as inferred in Fig 5B began to drop. We now point this out on p.8 lines 52-53 and p.9 lines 6-7. Previous work has shown clearly that during most of interphase, cyclin levels and Cdk1 activity rise linearly with time, and the prevailing hypothesis is that increasing concentrations of cyclin-Cdk1 in its interphase phosphorylation state are responsible for this linear rise in Cdk1 activity. See for example Pomerening *et al.* (Cell 2005) for a particularly detailed time course. In addition, the other relevant regulators of Cdk—Cdc25C and Wee1—do not change in their abundance or phosphorylation, and presumably in their activity, during this period. See for example Figure 7 in Kamenz *et al.* (Curr Biol 2021). These proteins only flip to their mitotic states immediately before mitotic entry when Cdk1 activity spikes up. We have unpublished data indicating that the same is true for Myt1. This supports the hypothesis that the approximately linear increase in Cdk1 activity shown in Fig 5C is due to cyclin synthesis rather than a change in the activity of other Cdk1 regulators. We now point this out explicitly on p. 8, lines 24-28.

I am also confused on how to reconcile the data on the duration of falling phase in Figure 4E, the data on cyclin degradation in Figure 5B and the dependency of APC and PP2A activity on temperature. The dependencies on temperature are very different, how do I square all this?

Response: We now address this question more explicitly in the revised manuscript. The differences in apparent temperature dependencies across Figures 4E, 5B, and the activity assays can be reconciled by considering three key points.

First, although Figure 5B suggests that the degradation rate changes relatively little with temperature, our model can still capture the observed changes in oscillation period if degradation is more temperature-sensitive. This means that the fitted values of these activation energies is not tightly constrained by the data. Specifically, the model remains consistent with the data as long as the cyclin synthesis rate increases more steeply with temperature than the degradation rate. This is supported by simulation results and by the ABC-inferred parameter distributions (Figures S10 and S11), which show that the apparent activation energy for synthesis, $E_a(k_s)$, is centered around 113kJ/mol. In contrast, $E_a(k_d)$ for degradation is very broadly distributed, and although the peak is at around 12kJ/mol, the mean is closer to 49kJ/mol (see also Fig.1 here — top middle panels of

Figure S10). This suggests that a range of degradation temperature sensitivities is compatible with the data, provided synthesis remains more sensitive. We have clarified this point in the revised text (p. 8, lines 10-22).

Second, our model is intentionally minimal (a two-variable ODE system) designed to extract mechanistic insight without overfitting. The real cell cycle regulatory network includes many additional components and interactions, and these likely contribute to the finer details observed in experiments. While our model captures the main features of the dynamics, explaining some differences in temperature responses (e.g., between degradation rate and APC/PP2A activity) would require a more detailed biochemical model. We discuss such model limitations in the text (p. 4–6).

Third, although APC/C activation is necessary to initiate cyclin degradation, it is not the sole determinant of degradation kinetics. After ubiquitination by APC/C, several downstream processes affect the actual rate of cyclin degradation: E2 enzymes extend ubiquitin chains; deubiquitinases can remove or edit them; the proteasome’s capacity and selectivity also play a role. In addition, post-translational modifications and subcellular localization of substrates influence degradation efficiency. Each of these steps could have its own temperature sensitivity, which helps explain why the overall degradation rate may not mirror the temperature dependence of APC/C or PP2A activity alone.

Figure 1: Marginal distributions of the scaling parameters of cyclin synthesis rate and the cyclin degradation rate for the optimal fits to extract data. These parameters determine the scaling of k_s and k_d shown in Fig. 4 and Fig S10. The marginal distribution over 1000 weighted samples, that are the result of the ABC algorithm, is shown. Smooth distribution obtained by Gaussian kernel density. Black line indicates the mean. The ‘Best fit’ quoted corresponds to the value of the parameter for the sample with least distance to the data.

2. The prediction for Figure 7A suggests a much bigger effect of reduction of cyclin synthesis

than it is experimentally observed. In the model, oscillations would stop before the cycle reaches a duration of ≈ 300 min, while in experiments oscillations of about 300 minutes are observed from 20-30 C when 6 μ M of Morpholinos are used. This links to the worry that cyclin synthesis is not the main or only determinant of the increase in Cdk activity. I think that the fact that there is an effect is interesting, but the authors are over-interpreting their results. Similarly in Figure 7C the gymnastic used to get a good fit of the model is not fully justified and the fit is actually not great with systematic deviations which are not properly acknowledged. Overall, I feel that the evidence for the biphasic nature of cyclin synthesis is weaker than the authors claim. That does not significantly impact my evaluation of the paper, but the authors need to present a more balanced conclusion. I would suggest that the author say that there is a biphasic dependency on temperature of the rate of Cdk activation in interphase and that the rate is partially but not fully controlled by the rate of cyclin synthesis.

Response: Our goal was not to achieve perfect quantitative fits but rather to use a minimal two-variable ODE model to extract broad mechanistic insights. Despite its simplicity, we find it striking that the model qualitatively captures many key features of the data, including the effect of reduced cyclin synthesis. For Figure 7A, we have revisited the simulations (see updated Figure 2A) and improved the fits, which now align much better with the experimental observations. We acknowledge that some discrepancies remain, possibly due to additional regulatory mechanisms beyond cyclin synthesis/degradation and Cdk1 activation, which are not included in the minimal model. Moreover, the extract system gradually slows over time (see Fig. 4B), whereas our simulations assume time-invariant parameters, which could explain part of the mismatch. Nonetheless, the key qualitative result — that increasing morpholino concentrations lead to longer minimum cycle periods and a narrower viable temperature range — is consistently captured by both model and experiments. We also now explicitly emphasize in the manuscript that Cdk activation is only partially determined by cyclin synthesis.

For Figure 7C (now updated as Figure 7B), we applied the same parameter set from Figure 5 to the extract data, making only minimal adjustments by uniformly scaling the cyclin synthesis and degradation rates at the reference temperature. This was motivated by the empirical observation that extract oscillations are systematically slower than embryo oscillations and allowed us to avoid extensive parameter tuning while preserving model transparency. Although the fit is not perfect,

we believe it reasonably captures the main trends, given the simplicity of the framework. To address the reviewer's concerns, we have reworked the layout of Figure 7 (see revised Figure 2) and added a new supplemental figure (Fig. S18) discussing the details of the rate scaling. We also revised the main text to more clearly acknowledge the limitations of the fit and the observed systematic deviations. Importantly, we have softened our conclusions: we now state that the rate of Cdk activation during interphase shows a biphasic temperature dependence and is influenced — but not solely determined — by cyclin synthesis. We believe this more balanced interpretation better reflects both the modeling and experimental evidence.

Major comments:

3. While I think that it is the authors' prerogative to write the paper they want, I did not enjoy reading this paper, as there was a lot of text explaining things that are either very basic or irrelevant to the flow. Just few examples: pages 3-5 have lengthy explanations of Arrhenius' law, a lengthy and pedantic explanation of Q10 which is disconnected from the rest of the paper, a lengthy explanation of how to fit double exponentials which could be moved to the methods (also I highly doubt that this is a paper where the amount of data required efficient fitting strategies and fitting two exponentials to data is not a computationally intensive procedure).

Response: We appreciate this feedback and understand the reviewer's concerns about flow and content prioritization. In response, we have revised the early sections of the manuscript to improve readability and streamline the narrative. Specifically, we have greatly shortened the explanations of Q10 and alternative fitting functions and moved the more technical details of exponential fitting to the Methods section. While we initially included these discussions to provide context for a broader audience, we recognize the importance of maintaining focus and have adjusted the text accordingly.

4. The extensions of the Arrhenius law and the fact that they fit data better is somewhat trivial as all the extensions are models with extra parameters and there is no attempt to correct for that. Furthermore, the insights from those fits are not relevant to understand the rest of the paper, so I found that the whole section adds little value to the paper and could be removed. To be clear, I feel that the authors should have the freedom to write the paper that they want to write and not the

one that I would write, so it is up to them whether to follow this suggestion and the previous one. However, I want to make the authors aware that, as a reader, by the time I got to the insightful stuff I had some fatigue from having to read through things that did not seem very insightful or relevant.

Response: See also point 3: we have reduced this section to one short paragraph to be more concise and focused. We now emphasize only the key point: that although multiple more complex functional forms fit the data better than the simple Arrhenius law, they all perform similarly well. This observation is used not to argue for a particular fitting function, but to highlight the limitations of purely empirical fitting in the absence of mechanistic models. This serves to motivate the modeling section that follows. We believe this restructuring improves the flow and makes the transition to the core contributions of the paper clearer and more engaging for readers.

5. The comments on *Drosophila* on page 4 are wrong. The cell cycle data the authors analyze is cycle 13 and there is no G2 in cycle 13. It is a longer cell cycle due to lengthening of S-phase but no G2 phase yet. Cycle 14 is asynchronous and lasts at least 75 min at room temperature. Similar data with temperature scaling at cycle 13 and with Cdk1 measurements are also present in Hayden et al *Current Biology* 2022 and could be cited and added to the authors' analysis.

Response: Thank you for pointing this out, we have now clarified this accordingly, and have added the citation.

6. The authors present two models for Cdk1 oscillations: a model dominated by positive feedback via Cdk1/Wee1/Cdc25 and a model dominated by the Cdk1/Greatwall/PP2A feedback. They then show that similar conclusions can be obtained on the dependency of temperature, that is the observed data require a biphasic dependency of synthesis and different temperature dependency. Interestingly, they also comment that the second model applies to cycle 2-12 *in vivo*, while the first model applies to the first cycle and *in vitro* cycles (I am extrapolating here, so please be clearer about which regimes the two models describe for the readers' information). It is interesting that the main "theoretical" conclusion of the paper is independent of the model used, but I also wonder whether modeling is even required and whether one could have reached this conclusion in a

model-independent manner, as it seems that modeling adds very little in a way. This is not a major concern to me, as I think that the main strength of this paper is in the experimental analysis which can be rationalized in a model-independent manner. This is just to say that I found the modeling part not very insightful. Again up to the authors to decide what to make of my comment here.

Response: We have now clarified the differences between the cell cycle models in more detail as follows: We asked whether the results obtained were specific to the two-ODE cell cycle model. To explore this, we turned to a structurally distinct model: a five-ODE, mass-action-based system that includes interactions among Cdk1, Greatwall, and PP2A—elements that collectively form a mitotic switch as well (62, 67, 74). In contrast to the two-ODE model, which featured a bistable switch between Cdk1 and cyclin B and included feedback through Cdc25 and Wee1, this model implements a bistable switch between APC/C and Cdk1, and thus omits the Cdc25/Wee1-mediated feedback loops entirely. It also differs in its use of strictly mass-action kinetics, avoiding the highly nonlinear Hill functions of the two-ODE model, and in its dimensionality, expanding from two to five ODEs. Despite these structural differences, both models share key features: cyclin synthesis and Cdk1-activated degradation, the presence of a bistable switch, and a separation of timescales enabling relaxation oscillations. The five-ODE model could be parameterized to yield realistic cell cycle oscillations (75) (Fig. S6E, Supplementary Note 3D). Due to the model’s increased complexity—ten kinetic parameters—we relied exclusively on the ABC algorithm for parameter inference. This approach produced satisfactory fits (Fig. S5C), and analysis revealed that highly correlated activation energy pairs typically corresponded to antagonistic reaction rates (see Supplementary Note D). These results again show that well-fitting parameter sets tend to exhibit similar activation energies for faster reactions. Moreover, they support the idea that thermal limits can arise from imbalances in the apparent activation energies of cyclin synthesis and degradation, reinforcing the conclusions drawn from the two-ODE model.

Regarding the role of modeling, we agree that the experimental data are compelling on their own, and we appreciate the reviewer’s recognition of the experimental strength of the paper. That said, we believe the modeling is both central and crucial to our work. It is uncommon that mechanisms of temperature scaling in biological oscillators are both proposed mechanistically and validated experimentally. The scaling curves that emerge from our models are not simple empirical fits—they

are nontrivial emergent features of the underlying dynamical systems, arising from the interplay between temperature-dependent synthesis and degradation rates. This highlights a broader conceptual point: temperature scaling need not rely on a single rate-limiting step or protein denaturation, as is often assumed, but can emerge from systems-level regulation. We now stress this point more clearly in the revised manuscript, as it may represent a shift in how temperature sensitivity in biological systems is understood. The modeling was thus essential for revealing that multiple mechanistic models converge on the same system-level behavior, and for showing how subtle temperature-dependent effects can produce complex scaling laws.

7. I don't find Figure 4E very helpful. What is the point of defining an activation energy for a system that clearly does not behave like Arrhenius? What does activation energy even mean then? Since the authors use activation energies later in Figure 6 for related molecular events, it is also a bit unclear how the data in Figure 4E should be interpreted. Maybe I missed something, but I was confused- see also my first point.

Response: We agree that referring to an "activation energy" in the context of a complex, multistep process such as the cell cycle can be misleading. In this case, the activation energy shown in Figure 4E was intended as a local apparent activation energy, capturing the temperature sensitivity of the process over small intervals—a conceptually similar measure to Q10. To reduce confusion and improve interpretability, we have now revised Figure 4E to show the local Q10 instead, in the same format as Figure 2D.

8. The analysis of Cdk activity and the additional biochemical experiments to get the different rates and their dependency on temperature is very interesting and the major contribution of this paper. The only minor comment I have is that the use of the Cdk1 sensor was inspired by work from Pines who made the first Cdk sensor and work from Di Talia who demonstrated the first use of the sensor in embryos, yet none of those papers are cited and only the paper from the Yang's lab that made the sensor specifically used here is cited. I think that it would be best to cite the papers that inspired making this tool in *Xenopus*.

Response: We have now cited the suggested works.

Figure 2: **Decreasing the cyclin synthesis rate decreases the viable temperature range**
 A. Influence of changing the basal cyclin synthesis rate by a factor up to 5 on the shape of the temperature response curves. The degradation rate is scaled up to a factor of 3. The two left panels show simulations of the 2-ODE model of the cell cycle oscillator using a parameter set obtained from the ABC method (one of the gray lines in Fig. 5B), plotting the temperature-dependence of the cyclin synthesis rate and the corresponding cell cycle period. The right panel shows the cell cycle duration as a function of temperature obtained from encapsulated extracts with 0, 4, or 6 μM morpholino (MO) oligonucleotides against isoforms of *Xenopus* cyclin B1/B2 mRNA species, thus lowering the cyclin synthesis rate. B. Different scenarios in temperature dependence of cyclin synthesis and degradation lead to different non-Arrhenius scaling of cell cycle oscillations. While a biphasic cyclin synthesis rate leads to a double exponential response curve, the imbalance in activation energies introduces a curved non-Arrhenius response at lower temperatures, which is critical for reproducing the experimental data measured in frog egg extract.

Reviewer #2 (Remarks to the Author):

This manuscript investigates the temperature-dependence of the *Xenopus laevis* cell cycle at various levels by experiments in fertilized oocytes, oocyte extracts, and by mathematical/computational modeling with ODEs and nonlinear dynamics. Valuable comparisons with fish and worm embryos provide insight into the generality of the observations. The findings indicate Arrhenius scaling over a range of temperatures for embryo cell division timings and other post-fertilization events. While the actual timings may be different, the activation energies are remarkably similar across events and species. A simple and elegant mathematical model leads to a relaxation oscillator that matches the essence of embryo cell cycle dynamics with three-phase periods. Examining the 4 rates in the model leads to the conclusion that either independent temperature-scaling or biphasic scaling of at least one parameter's temperature-dependence explains the deviations in cell cycle timings from the Arrhenius model outside of the good-fit temperature range. Further experiments highlight that both independent and biphasic scaling are at play, with cyclin synthesis being more temperature-sensitive than cyclin degradation, and differences in the temperature-scaling of interphase and M-phase durations.

This is a great interdisciplinary manuscript that was a joy to read both from a theory and experimental perspective. It provides many significant insights into fundamental questions that also have environmental relevance in a changing climate. The work is well executed and supports the conclusions. I would like to recommend publication after the authors address the following comments.

We thank the reviewer for this generous and encouraging feedback. We are pleased to hear that the interdisciplinary approach and the integration of theory and experiment is compelling. It is also rewarding to know that the broader relevance of our findings came across clearly. We appreciate the thoughtful comments and address them carefully here below and in the revised manuscript.

1. The investigation of similarities between different animal species gives nice insights. It would be interesting to expand more. Since yeast is mentioned, how would these results generalize to eukaryotes? Previous work showed a cell fate decision at high temperatures in yeast, which would be worth discussing, see PMID:30341217. Another interesting reference might be on the temperature-

compensation of a synthetic bacterial oscillator, see PMID:24395809.

Response: This is an interesting question and thank you for sharing these references. We have now included a discussion of these two relevant references (PMID: 30341217 and PMID: 24395809) in the main text as future directions for expanding the scope of our approach (**To do still**). These studies highlight how temperature-induced changes in gene expression and protein conformation can significantly affect gene regulatory network dynamics. For example, the work by Charlebois et al. (PMID: 30341217) demonstrates that non-optimal temperatures can induce a cell fate decision between growth arrest and stress resistance in yeast, dramatically affecting gene expression patterns. This behavior was shown to arise from multiple temperature-sensitive layers, including reaction kinetics, growth rates, and protein dynamics. Hussain et al. (PMID: 24395809) engineered a synthetic bacterial oscillator that achieves temperature compensation through a protein-level modification, a single amino acid mutation in a transcriptional repressor, highlighting how tuning molecular properties can buffer against Arrhenius scaling.

Our current study differs in its focus on the early embryonic cell cycle, a system that is (largely) transcriptionally silent and lacks checkpoint controls. This simplifies the regulatory network and allows us to study temperature effects primarily at the level of protein interactions and enzymatic activity, without the added complexity of gene expression regulation. In this context, our key finding is that temperature scaling properties can emerge from the structure and dynamics of the network itself, without requiring a built-in molecular temperature switch such as a conformational protein change or specific mRNA splicing mechanism. This offers a complementary perspective: rather than relying on dedicated components that actively compensate for temperature changes, scaling behavior can be a collective emergent property of the biochemical network. By extending this modeling framework to systems with gene expression, we believe it will be possible to uncover general principles of temperature compensation, particularly how network architecture buffers thermal effects without fine-tuning individual reactions.

2. As exposed in PMID:30341217, temperature affects cells at multiple scales, including gene expression and protein structure. It would be interesting to discuss how gene expression alterations due to the temperature-dependence of mRNA and protein synthesis and degradation/dilution for the cell cycle machinery, including Cdk1, Cdc25, Wee1 could affect the model – which is mostly at

the protein interaction/activity level.

Response: This is an interesting question. In this study, we focus on the early embryonic cell cycle, where our modeling approach is applicable because gene expression is still largely silenced. Beyond this stage, the system becomes considerably more complex, making it much more challenging to set up a dynamical model of the regulatory network. Recent theoretical work (Voits and Schwarz, arXiv, 2024) suggests that large biochemical networks can give rise to quadratic exponential scaling, where deviations from Arrhenius behavior emerge from directional biases and differing activation energies. This aligns well with our observation that cyclin synthesis (a forward process) and cyclin degradation (a backward process) have distinct temperature sensitivities. It would indeed be valuable in future work to test these ideas experimentally, and to explore how different network architectures—including those involving gene expression—affect temperature responses.

3. In Fig. 1, how does the period between fertilization and fertilization wave completion depend on temperature? It would be interesting to know, since the other periods are all plotted.

Response: Thank you for this suggestion. We indeed think that it is more informative to plot the durations between sequential events in the main figure. Accordingly, we now include the following intervals in the new Fig. 1 (see Fig. 3 below): (1) fertilization to initiation of the fertilization wave; (2) fertilization wave to surface contraction wave; (3) surface contraction wave to first cleavage; and (4) pooled durations for cell cycles from the 2nd to 4th cleavage. The remaining durations, including those shown previously, are now presented in the supplemental material. We have updated the main text to reflect this change. Parts of the previous Figure 1 are now moved to Fig. S1.

4. Unlike the rest of the paper, the last section before the Discussion is somewhat unclear. Can this be more clearly written? What are morpholino oligonucleotides and how do they help address the question in this section?

Response: Thank you for this feedback. We have revisited this section and polished the writing for greater clarity. To clarify: morpholino oligonucleotides are synthetic molecules that bind

specifically to mRNA transcripts and block their translation without degrading the RNA. In our experiments, we used morpholinos targeted against cyclin B mRNA to reduce the effective rate of cyclin B protein synthesis. This allowed us to directly test the prediction from our model that decreasing cyclin synthesis should reduce the temperature range over which cell cycle oscillations are viable.

This section of the manuscript investigates how the temperature limits of the *Xenopus* embryonic cell cycle oscillator arise from two distinct mechanisms. At low temperatures, failure is due to an imbalance in the temperature dependence of synthesis and degradation rates; at high temperatures, failure stems from a biphasic cyclin synthesis response that drops off with increasing temperature. Using both modeling and morpholino perturbations, we show that reducing cyclin synthesis narrows the viable temperature range, consistent with model predictions. Furthermore, we demonstrate that our minimal model calibrated to extract data can reproduce large parts of the embryo temperature response with only modest adjustments. Taken together, these results support a mechanistic interpretation in which both the Arrhenius imbalance and the biphasic nature of cyclin synthesis are essential to explain the observed non-Arrhenius scaling behavior of the early embryonic oscillator. We have also adjusted the corresponding Figure 7 (here Fig. 2 above) to increase clarity.

5. How would these temperature-dependences act in adult animals with full cell cycle and developed tissues? Any implications for human biology? Generality for genetic oscillators? There are interesting and possibly relevant observations for p53 oscillations, see PMID:32001771.

Response: This is indeed an important question. While we do not have a clear answer, we can offer several thoughts.

Our study focuses on early embryos and simplified in vitro systems, which differ substantially from adult organisms with fully developed tissues. Modeling these mature systems in detail is beyond our current scope, but studying subsystems or general features using coarse-grained models is a promising path forward.

With respect to human biology, it is particularly interesting to consider how the behaviors we observe might compare to those of similar circuits in endotherms, organisms that, at considerable metabolic cost, maintain constant body temperatures. This regulation frees their biochemical systems from the need to operate robustly over wide temperature ranges. As we note in the discussion, although

the four enzymatic processes we analyzed—cyclin synthesis, cyclin degradation, Cdk1 activity, and PP2A-B55 activity—differed in their temperature sensitivities, the differences were relatively modest: their apparent activation energies (E_a) averaged about 60 kJ/mol, with a standard deviation of 16 kJ/mol (roughly 27%). This suggests that, in ectotherms, oscillator components may need to be more tightly coordinated in their temperature scaling to maintain functional rhythms. In contrast, endothermy might permit greater diversity in enzymatic activation energies, relaxing this constraint.

The reference to p53 oscillations (PMID:32001771) is also well-taken. These oscillations represent a critical genetic circuit in human cells, especially in the context of DNA damage response. It would be fascinating to explore whether the temperature-scaling principles identified here might also apply to such stress-responsive oscillators (or transient dynamics) in endotherms. While our current work does not address this directly, it points toward valuable directions for future research.

More broadly, we believe that the principles uncovered here, particularly the way in which production and degradation terms interact in temperature-dependent relaxation oscillators, may be generalizable. Many biological oscillators are based on negative feedback with time scale separation, and we expect similar scaling behaviors to emerge in such systems. Our findings may therefore reflect a broader design principle relevant across diverse biological contexts, even beyond early development.

6. Are there any clues whether the imbalanced temperature-sensitivity of cyclin synthesis versus degradation could be conserved in other animals, or even yeast?

Response: This is an intriguing idea. We believe that the concept, that an imbalance in the temperature sensitivities of forward versus backward processes can lead to non-Arrhenius scaling, is a general one. In the early embryonic cell cycle, the key opposing processes are cyclin synthesis and degradation. However, in later developmental stages or in the cell cycles of adult organisms, the relevant processes may involve a more complex interplay of multiple forward and reverse reactions. This kind of temperature-dependent asymmetry requires a system with clear directionality, such as a well-defined progression through cell cycle phases or regulatory cascades. It would be interesting to investigate whether similar imbalances exist in other organisms, including yeast, and whether they produce comparable deviations from simple Arrhenius behavior.

7. What are the implications of these findings on designing or evolving synthetic genetic oscillators? A relevant reference would be PMID:38925113 when discussing this aspect.

Response: Our findings suggest that temperature-dependent behavior in biological oscillators can arise from imbalances in the activation energies of opposing processes, such as cyclin synthesis and degradation. This differential temperature sensitivity may apply broadly to synthetic genetic oscillators with negative feedback. The shared reference (PMID:38925113) on mid-scale evolution offers a promising framework to explore this idea further. By evolving entire gene circuits under controlled, fluctuating temperature conditions, it may be possible to select for oscillators that are either robust across a temperature range or finely tuned to specific thermal regimes. Such strategies would evolve circuits with coordinated temperature scaling across the network, as seen in ectothermic systems adapted to environmental variability. Conversely, circuits operating in thermally stable environments—such as in endothermic organisms or controlled bioreactors—may tolerate or even benefit from broader variation in activation energies. We believe that evolving synthetic networks under defined temperature regimes could provide valuable insights into how cells maintain robust oscillatory dynamics in the face of thermal fluctuations. These insights could inform both the rational design and adaptive evolution of synthetic oscillators with predictable, tunable, and environmentally resilient behavior. We have now added a concise discussion of these ideas to the main text.

8. Figure 2B: Maybe indicate the optima by an arrowhead or some other symbol.

Response: Done.

Reviewer #2 (Remarks on code availability):

Unfortunately access to the code is restricted. It seems like only KU Leuven faculty or staff members or students can access the code.

Response: Our apologies. We were not aware that access was previously limited to KU Leuven

staff, but this has now been corrected, and the code is fully publicly accessible.

Figure 3: Cell division timing in early *Xenopus laevis* embryos scales approximately Arrhenius over a wide range of temperatures. A. *Xenopus laevis* embryonic development was imaged in a temperature-controlled chamber. The time unit mpf is minutes post-fertilization. B. Different early developmental events were visually identified. SEP denotes the sperm entry point. C. Duration of several early developmental periods in function of temperature in the range [$T_{\min} = 9\text{C}, T_{\max} = 29\text{C}$]. D. An Arrhenius fit is shown for the values between 12°C and 21°C, with the apparent activation energy indicated. E. Bootstrapping provides a probability distribution for the apparent activation energies. The mean and 90% confidence interval (CI) are also indicated.